# Avalanche danger level characteristics from field observations of snow instability

Jürg Schweizer[1], Christoph Mitterer[2], Benjamin Reuter[3], Frank Techel[1]

[1] WSL Institute for Snow and Avalanche Research SLF, Davos, Switzerland
[2] Avalanche Forecasting Service Tyrol, Innsbruck, Austria
[3] Centre d'Etude de la Neige, CEN, CNRM, MétéoFrance, Grenoble, France

*Correspondence to*: Jürg Schweizer (schweizer@slf.ch)

**Abstract.** Avalanche danger levels are described in qualitative terms that mostly are not amenable to measurements or observations. However, estimating and improving forecast consistency and accuracy requires descriptors that can be observed
or measured. Therefore, we aim to characterize the avalanche danger levels based on expert field observations of snow instability. We analyzed 589 field observations by experienced researchers and forecasters recorded mostly in the region of Davos (Switzerland) during 18 winter seasons (2001-2002 to 2018-2019). The data include a snow profile with a stability test (rutschblock, RB) and observations on snow surface quality, drifting snow, signs of instability and avalanche activity. In addition, observers provided their estimate of the local avalanche danger level. A snow stability class (*very poor*, *poor*, *fair*,
*good*, *very good*) was assigned to each profile based on RB score, RB release type and snowpack characteristics. First, we describe some of the key snowpack characteristics of the data set. In most cases, the failure layer included persistent grain types, even after a recent snowfall. We then related snow instability data to the local avalanche danger level. For the danger levels *1–Low* to *4–High*, we derived typical stability distributions. The proportions of profiles rated *poor* and *very poor* clearly increased with increasing danger level. For our data set, the proportions were 5 %, 13 %, 49 % and 63 % for the danger levels
*1–Low* to *4–High*, respectively. Furthermore, we related the local avalanche danger level to the occurrence of signs of instability such as whumpfs, shooting cracks and recent avalanches. The absence of signs of instability was most closely related to *1–Low*, the presence to *3–Considerable.* Adding the snow stability class and the 3-day sum of new snow depth improved the discrimination between the lower three danger levels. Still, *2–Moderate* was not well described. Nevertheless, we propose some typical situations that approximately characterize each of the danger levels. Obviously, there is no single easily
observable set of parameters that would allow fully characterizing the avalanche danger levels. One reason for this shortcoming is the fact that the snow instability data we analyzed usually lack information on spatial frequency, which is needed to reliably assess the danger level.

# 1    Introduction

Snow avalanches in populated snow-covered mountain regions may threaten people and infrastructure. To mitigate the effects of avalanches, among other things, warnings are issued by avalanche forecasting services that inform the public about when and where the danger is most imminent. The danger is described with the help of a danger scale including five levels: *1–Low, 2–Moderate, 3–Considerable, 4–High* and *5–Very High* (or Extreme) (e.g., EAWS, 2021; Statham et al., 2018). The danger levels were originally characterized by the avalanche release probability (which increases if stability decreases), the spatial distribution of snow instability and the avalanche size (Meister, 1995). These three components are assumed to increase with increasing avalanche danger (level). Forecasts are issued at the regional scale for areas typically larger than 100 km$^2$, i.e. they do not refer to single slopes or endangered objects (infrastructure), which need to be assessed based on the site-specific conditions (e.g., Reuter and Semmel, 2018; Stoffel and Schweizer, 2008). Hence, the danger level describes the probability of natural or skier-triggered avalanches and the possible extent of avalanches in a particular region (Föhn, 1994; Meister, 1995).

However, the avalanche danger that exists in an area cannot be measured – at best, it can be estimated (Föhn and Schweizer, 1995; Giraud et al., 1987). Therefore, forecast verification is not straightforward; even in hindsight, evidence of danger such as avalanches may be lacking. Nevertheless, avalanche activity, or indices describing it, has been used to verify the forecast (Giraud et al., 1987). However, as several verification studies pointed out, this approach is not feasible for the danger levels *1–Low, 2–Moderate* and *3–Considerable* (e.g., Föhn and Schweizer, 1995). Schweizer et al. (2003) therefore used point snow instability data instead and characterized the lower three danger levels by stability distributions. On the other hand, the higher two danger levels are clearly characterized by numerous natural avalanches (e.g., Schweizer et al., 2020).

Alternatively, estimates of local avalanche danger (local nowcasts) by experienced experts can be used to assess the accuracy of regional forecasts (e.g., Harvey et al., 1998; Jamieson et al., 2009; Techel and Schweizer, 2017). These local nowcasts are based on observations recorded during a day travelling in the backcountry and typically refer to a smaller area within the region covered by the public forecast, but do not refer to a single slope. Among the most indicative observations to assess snow instability and hence the local danger are signs of instability such as whumpf sounds, shooting cracks and recent avalanching as well as snow stratigraphy observations and snow instability tests (e.g., Schweizer and Jamieson, 2010).

However, there is no specific set of observations that unequivocally relates local observations to one of the lower three danger levels. Simple field observations help to downscale from a regional to a local danger level (Jamieson et al., 2009). Their classification tree analysis suggests that on a particular day with regional forecast *3–Considerable* and at least one sign of instability observed, either recent slab avalanching, shooting cracks or whumpfs, the local danger estimate was in fact *3–Considerable* – otherwise it was *2–Moderate*. A similar study by Bakermans et al. (2010) using snowpack observations revealed the usefulness of the avalanche bulletin. Stability test variables were correlated with the local danger level estimate, yet the strongest correlation they found was still between the regional forecast and the local danger estimate. Schweizer (2010) preliminarily analyzed whether the danger level could be inferred from signs of instability. Shooting cracks were mostly

associated with the danger level *3–Considerable*, whereas in the absence of any signs of instability the danger level was lower, but it was not possible to distinguish *2–Moderate* from *1–Low*. Two more recent studies tried to quantify the danger levels based on stability test data and observed avalanche activity. Techel et al. (2020) related snowpack stability, the frequency distribution of snowpack stability and avalanche size to the avalanche danger levels *1–Low* to *4–High*. The best predictor was the frequency of profiles rated as very poor, followed by the maximum avalanche size. Avalanche activity and avalanche size were related to the danger levels by Schweizer et al. (2020). With increasing danger level, avalanche activity strongly increased, while avalanche size did not generally increase, in particular not at the lower danger levels (*1–Low* to *3–Considerable*). Only at danger levels *4–High* and *5–Very High*, the proportion of large and very large avalanches clearly increased.

While avalanche danger is obviously best described in terms of snow stability, the frequency distribution of snow stability and avalanche size (Techel et al., 2020), comprehensive measurements of this set of quantities are virtually nonexistent. Instead proxies are used such as signs of instability to assess the avalanche danger – though it is currently unclear how useful these proxies are. Hence, the aim of the present study is to further foster the quantitative description of the avalanche danger levels, by characterizing the danger levels based on proxies, namely field observations of snow instability. To this end, we analyze a data set of 589 snow profiles with a snow instability test (rutschblock) and additional information. These data include observations on signs of instability and an estimate of avalanche danger (local nowcast) for the date and area of the profile.

## 2      Data and methods

We analyzed a data set of 589 snow profiles that were almost exclusively (95 %) observed in the region of Davos (Eastern Swiss Alps) during the winter seasons of 2001-2002 to 2018-2019 (18 winters). As of winter 2001-2002 observers were asked to complement snow profile records with field observations to support their interpretation in operational forecasting. These observations include specific, additional information related to the snow profile, namely (1) profile site, (2) snow cover characteristics, (3) stability test results, as well as other relevant observations made while travelling the backcountry, on (4) snow surface properties, (5) drifting snow, (6) whumpfs and shooting cracks, and (7) recent avalanche activity; these observations are completed with (8) an estimate of the local avalanche danger (local nowcast: LN). The local nowcast refers to the area where the observers made their observations; this area is smaller than, but part of, the region for which the regional danger level was forecast (regional forecast: RF). Two observation examples are shown in Table 1.

We selected 589 profiles from SLF's snow profile database based on whether (a) all text fields had complete information (Tab. 1), (b) an experienced observer had done the pit and (c) snow conditions were dry, since only for dry-snow conditions the field observations we analyze are indicative of avalanche danger. We considered information to be complete if, for instance, in the category "signs of instability and recent avalanching" not only the presence but also the absence of these observations

was explicitly stated. We considered observers as experienced if they had done dozens of high-quality profiles. Almost all (96 %) were either professional forecasters or researcher with extensive experience in field work; the authors recorded 63 % of all profiles.

Table 1: Two examples of field data collected in the course of a snow profile observation.

| Data entry | Example 1: Salezer Mäder, 24 Feb 2015 | Example 2: Latschüel, 23 Feb 2017 |
|---|---|---|
| (1) Profile site | On a small roll, lee slope. Snow depth above average. | In the lower half of medium-sized slope. Snow depth below average. Rocky ground. Representative for slopes of this aspect and elevation. |
| (2) Snow cover characteristics | Thick layer of new snow on hard thick crust, well consolidated due to sun. Basal layers seem slightly moist. | In general, poorly consolidated. Early winter depth layers are down about 70 cm. No prominent weak layers in the more recent layers. |
| (3) Stability test results | RB released when stepping onto it, but only below skis. CT and ECT show several failures in the new snow layers. | RB: only after several jumps one edge did break away: failure below the crust in depth hoar layer, down 70 cm. Other snow stability tests (ECT) showed similar results. |
| (4) Snow surface properties | About 20 cm of new snow, smooth. No new snow in wind exposed areas at all, but hard surface (crust on sunny aspects) or strongly faceted surface layers in shady aspects. | Mainly smooth, crusty on sunny slopes. |
| (5) Drifting snow | New small to medium-sized wind slabs are forming. | No recent snow drift accumulations observed, though occasionally some drifting snow in the morning. |
| (6) Signs of instability: whumpfs and shooting cracks | None observed. | No signs of instability observed. |
| (7) Recent avalanching | Small wind slab observed at Salezerhorn east ridge, 2100 m a.s.l.; triggered a small wind slab near Damm. | No recent slab avalanches observed. Many sluffs in steep sunny slopes. |
| (8) Local danger estimate (local nowcast) | Considerable above treeline. New snow drifts can be easily triggered, in all aspects, but fairly localized. | Moderate, above 2200 m a.s.l., in western, northern and eastern aspects. |

All profiles were recorded on slopes. The most frequent aspect was north; almost 60 % of all profiles were observed in slopes facing northwest, north or northeast. Slope angle varied between 14 and 43°; 75 % of the profile slopes were steeper than 30°, with a median of 33°. Median elevation was 2470 m a.s.l.; less than 10 % of the profiles were observed below tree line. Snow profile records included snow stratigraphy (grain type and size, snow hardness index; Fierz et al., 2009), snow temperature and a rutschblock test (RB; Föhn, 1987). For the rutschblock test the score (1 to 7), release type (whole block,

partial release below skis or only an edge) and quality of fracture plane (smooth, rough, irregular) were recorded (Schweizer and Jamieson, 2010).

      For the analysis, all profiles were classified into 10 hardness profile types (1 to 10) and five stability classes (1: *very poor*, 2: *poor*, 3: *fair*, 4: *good* and 5: *very good*) following Schweizer and Wiesinger (2001). This 5-class stability scheme (Appendix A) considers the RB test result (score and release type), failure layer properties, snow stratigraphy and snowpack

structure.

      For the layer and height where the RB failed, we calculated the stratigraphy-based threshold sum using six unweighted variables: difference in grain size, failure layer grain size, difference in hardness, failure layer hardness, failure layer grain type and failure layer depth. The threshold values (or critical ranges) we used were those suggested by Schweizer and Jamieson (2007). In case the RB failed at several locations in the profile, we selected the most relevant failure interface based on depth,

RB score and RB release type. In general, we chose the failure interface that had the lowest score. In case there were two failure interfaces with the same score, but different depth, we opted for the failure interface where the whole block released, otherwise for the one closer to the surface. Once the primary failure interface was found, we followed the approach that was introduced by Schweizer and Jamieson (2007). In most cases it was obvious which of the two layers across the failure interface was the failure layer; if not we selected the softer of the two layers as the failure layer (FL) and the layer across the failure

interface as the so-called adjacent layer (AL). If there was no difference in hardness, we selected the lower layer as the failure layer, and the layer above the failure interface as the adjacent layer. Differences in grain size and hand hardness index were calculated across the failure interface, i.e. between failure layer and adjacent layer.

      As suggested by Schweizer et al. (2008), we then combined the threshold sum, the RB score and RB release type to obtain a simple alternative stability classification (A: *POOR*, B: *FAIR*, C: *GOOD*). We refer to this classification as the 3-class

stability scheme.

      In addition, for each profile, the snowpack structure was classified by calculating the snowpack index $SNPK_{index}$ as suggested by Techel and Pielmeier (2014). This index includes three elements: properties of (1) the slab (thickness), (2) weakest layer interface and (3) the proportion of the snowpack that is soft, coarse-grained and consists of persistent grain types.

When analyzing the text fields, we simply considered the presence (1) or absence (0) of whumpfs, shooting cracks and recent avalanching. For recent avalanching, we did not consider sluffs, wet-snow or glide-snow avalanches, but only dry-snow slab avalanches. For the local nowcast, observers occasionally (in 14 % of the cases) provided intermediate values of the danger level, which for analysis were assigned to the next full danger level (e.g., 2- and 2+ to *2–Moderate*, or 2-3 to *3–Considerable*).

We also considered new snow depth for analysis. As these data were not recorded by the observers, we compiled snow depth data from measurement stations in the surroundings of the snow profile locations. We considered the median snow depth from all new snow measurements, either manually observed or calculated from automatic snow depth recordings at automatic weather stations (Lehning et al., 1999), within a radius of 10 km of the profile location. Measurement locations had to be within ± 300 m in elevation from the elevation of the profile location. Occasionally, in about 3 % of the cases, the radius had to be increased to 20 km.

Finally, the data set was completed with the most recent regional danger level as forecast in the public bulletin (regional forecast: RF) for the day of observation ($N = 583$). Hence in a few cases in early winter, the danger forecast was not available since operational forecasting had not started yet.

When analyzing the occurrence of signs of instability, we compare the relative frequency of e.g. whumpfs at danger level *1–Low* to the base rate, i.e. how often whumpfs were reported at all. Moreover, we provide the odds ratio. The calculation of the odds, which is the ratio of a probability to its complementary probability, $P /(1 - P)$ is based on the usual $2 \times 2$ contingency table used in forecast verification (Wilks, 2011; p. 307). The odds ratio $\theta$ is defined as the ratio of the conditional odds of a hit $a$, given that the event occurs $\frac{a}{b}$, to the conditional odds of a false alarm $b$, given that the event does not occur $\frac{c}{d}$:

$$\theta = \frac{a}{b} : \frac{c}{d} = \frac{a\,d}{b\,c} \qquad\qquad (1)$$

with $c$ the misses and $d$ the correct non–events. As we have four different danger levels we provide the odds ratio for each danger level compared to either the rest, e.g. *1–Low* vs. the group of *2–Moderate*, *3–Considerable* and *4–High*, or compared to the higher (or lower danger levels), e.g. *2–Moderate* vs. the group of *3–Considerable* and *4–High*. Also, we compare the odds between the lower two to the higher two danger levels: *1–Low* and *2–Moderate* vs. *3–Considerable* and *4–High*.

To assess the uncertainty of the observed stability distributions, we sampled 50 times from the observed distributions (bootstrap) and calculated the mean frequency and its standard deviation for each stability class. To compare distributions, we applied the non-parametric Mann-Whitney *U*-Test. We selected a level of significance $p = 0.05$ to judge whether the observed differences were significant. We also checked for equality of proportions in $2 \times 2$ contingency tables. Relations between variables were described with the Spearman rank-order correlation coefficient $r_{sp}$. To compare distributions of categorical variables data were cross-tabulated and the Pearson $\chi^2$ statistic was calculated (Spiegel and Stephens, 1999). Finally, we

evaluated whether we can relate the observed signs of instability to the danger levels with the help of a classification tree (Breiman et al., 1998).

Our data set was unbalanced with regard to the danger levels. Therefore, for the detailed, cross-validated classification tree analysis, we randomly sampled 131 cases for the danger levels *2–Moderate* and *3–Considerable*, which corresponds to the number of cases with *1–Lo*w, to create 10 balanced data sets with each 131 cases of the three lower danger levels (*4–High* not included).

## 3 Results

We will first present relevant results on the snowpack characteristics of our data set, then report on the frequency of snow stability classes in relation to the avalanche danger level and finally present the results on characterizing the danger levels based on the signs of instability, the snow stability class and new snow depth.

### 3.1 Snowpack characteristics

In the following, we present the characteristics of the snow profile data, the stability test results and thereof derived stability, and selected relations between these data.

#### 3.1.1 Snow depth

Snow depth at profile sites varied between 42 and 260 cm with a median of 108 cm, which is less than the mean snow depth at Weissfluhjoch (2540 m a.s.l.) within the period mid November to mid March: 133 cm. Hence, snow depth at the profile sites was below average compared to the Weissfluhjoch site. The latter we consider as reference, since most profiles were from the region of Davos: 90 % of the profile locations were less than 13 km from the Weissfluhjoch site.

#### 3.1.2 Failure layer characteristics

Characteristics of the failure interface, which are relevant for calculating the threshold sum, are summarized in Table 2. The median failure layer depth (or slab depth) was 38 cm, with the lower quartile at 25 cm and the upper at 52 cm. In 87 % of the profiles the failure layer depth was within the so-called critical range [18 cm, 94 cm] as defined in the threshold sum approach. The median failure layer thickness was 7.7 cm with the middle 50 % ranging from 3 to 15 cm.

Most failure layers, 484 out of 589, were very soft or soft with a hand hardness index of 1 (Fist) or 1 to 2 (Fist to 4 Fingers). In about three quarters of the cases the mean failure layer grain size was 1 mm or less.

In 417 out of 589 cases (71 %) the failure layer consisted of persistent grain types, with faceted crystals (FC) the most common type. In the remaining 172 cases, where the grain type was non-persistent, most failure layers included small rounded grains (RG). Failure layers with non-persistent grain types were mostly located above the failure interface (in 129 out of 172 cases; 75 %). In contrast, for failure layers with persistent grain types, the failure layer was usually below the failure interface

(in 270 out of 417 cases, 65 %). Overall, independent of grain type, 53 % of the failure layers were below the failure interface (Table 2).

Failure layers with persistent grains (median: 40 cm) were deeper in the snowpack than failure layers with non-persistent grains (median: 30 cm; Mann-Whitney $U$-Test, $p < 0.001$; Figure 1). Moreover, snow depth was lower in cases with failure layers consisting of persistent grain types (median: 102 cm) than of non-persistent grain types (median: 122 cm; Mann-Whitney $U$-Test, $p < 0.001$).

**Table 2: Characteristics of failure interface (FI) with properties of failure layer (FL) and adjacent layer (AL); FL grain types: faceted crystals (FC), rounded grains (RG), rounding faceted particles (FCxr); failure layer hardness is shown as hand hardness index: 1 (*Fist*), 2 (*4 Fingers*), 3 (*1 Finger*), 4 (*Pencil*); N = 589.**

| Property | Median or modus | 1st and 3rd quartile; 2nd and 3rd most frequent |
|---|---|---|
| Snow depth | 108 cm | 82 – 137 cm |
| FI depth | 38 cm | 25 – 52 cm |
| FL thickness | 7.7 cm | 3 – 15 cm |
| FL hardness index | 1 | 1 – (1 to 2) |
| FL grain type | FC | RG, FCxr |
| FL grain size | 1 – 1.5 mm | (0.5-1.5) – (1.0-2.5) mm |
| AL thickness | 5 cm | 1 – 13 cm |
| AL hardness index | 2 to 3 | 2 – (3 to 4) |
| AL grain size | 0.75 – 1 mm | (0.5-1.0) – (0.75-2.0) mm |
| Hardness difference across FI | 1 | 0.5 – 2 |
| Grain size difference across FI | 0.5 mm | 0.25 – 1 mm |
| FL above/below AL | 277 above, 312 below | - |
| Threshold sum (number of "lemons") | mode: 5 (median: 4) | 4, 3 |

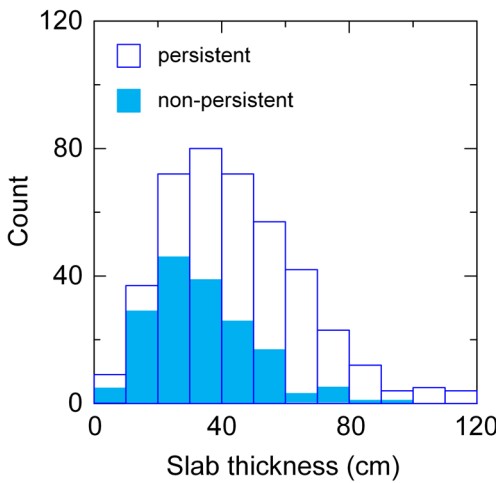

Figure 1: Frequency of slab thickness for failure layers with non-persistent grains (light blue; $N = 172$) and with persistent grains (white; $N = 417$).

### 3.1.3 New snow

In 29 % of the cases no new snow was recorded the day the profile was observed nor on the two preceding days (HN3d $\leq$ 1 cm; $N = 172$). In the other cases, the median 3-day sum of new snow depth was 13 cm ($N = 417$). In less than 10 % of these cases a major snowfall (HN3d $\geq$ 50 cm) preceded the day of observation.

With regard to failure layer depth and grain type in the failure layer (non-persistent vs. persistent) differences for days with and without new snow were small. Considering the cases without new snow (HN3d $\leq$ 1 cm), the failure layer depth was 200 again larger for the failure layers with persistent grain types (median: 39 cm) than for the layers with non-persistent grain types (median: 26 cm; Mann-Whitney $U$-Test, $p = 0.01$). For the cases with new snow, the median 3-day sum of new snow depth was 13 cm for the cases with persistent grain types in the failure layer and slightly larger, namely 16 cm, for the cases with non-persistent grain types in the failure layer; the difference was significant (Mann-Whitney $U$-test, $p = 0.018$).

For cases with HN3d $\leq$ 10 cm ($N = 349$) the proportion of cases with persistent failure layers was significantly larger 205 (74 %) compared to 66 % for cases with HN3d > 10 cm ($p = 0.028$). In the 41 cases with a substantial amount of new snow (HN3d $\geq$ 50 cm) the proportion of failure layers with persistent grain types was still 49 %, though significantly lower than the base rate of 71 % (proportion test, $p = 0.005$). Overall, persistent grain types were dominant in failure layers almost regardless of the amount of new snow.

### 3.1.4 Rutschblock test results

The RB scores 3 to 6 were all similarly frequent in the data set; the median RB score was 4. RB score increased with increasing failure layer depth ($r_{sp} = 0.28$, $p > 0.001$). The frequency of the release type *whole block* decreased with increasing RB score

(Figure 2). Its relative frequency at RB score 2 was 74 %, whereas it was only 5 % at RB score 6. This finding may be interpreted to mean that with increasing load needed for failure initiation also the propensity for crack propagation decreases.

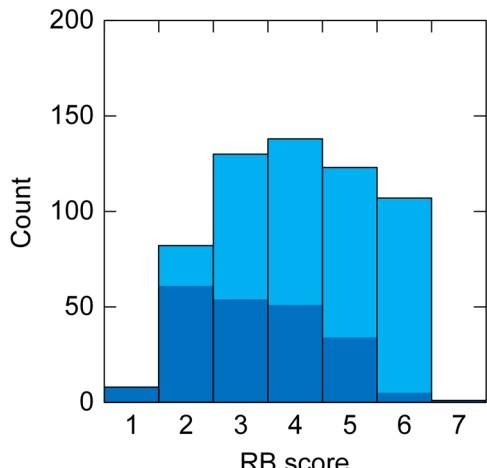

**Figure 2: Frequency of rutschblock scores (1 to 7; light blue; $N = 589$); dark blue bars highlight cases with release type *whole block*.**

In most of the RB tests (64 %), the quality of the fracture plane was characterized as smooth, in 21 % as rough and in 15 % as irregular. For tests with RB scores 1 to 3, the proportion of smooth fracture planes was slightly larger than average, and vice versa for tests with RB scores 4 to 6. On the other hand, the proportion of irregular fracture planes clearly increased with increasing RB score. Therefore, an irregular fracture plane indicates rather favourable conditions, whereas smooth or rough failures do not allow discrimination in terms of stability.

**3.1.5  Snow instability ratings**

The distribution of point snow stability estimated with the five class scheme according to Schweizer and Wiesinger (2001) was centered at 3: *fair* (Figure 3). More profiles were rated as *good* and *very good* (32 %) than *poor* and *very poor* (25 %; proportion test: $p = 0.003$). The proportion of profiles rated as *very poor* and *poor* was larger for failure layers with persistent grain types than for failure layers with non-persistent grain types (29 % vs. 15 %, p = 0.001). Among the *poor* and *very poor*
profiles, the majority (86 %) were cases with RB score $\leq 3$, *whole block* release and failure layer threshold sum $\geq 5$, which are criteria considered in the stability assessment.

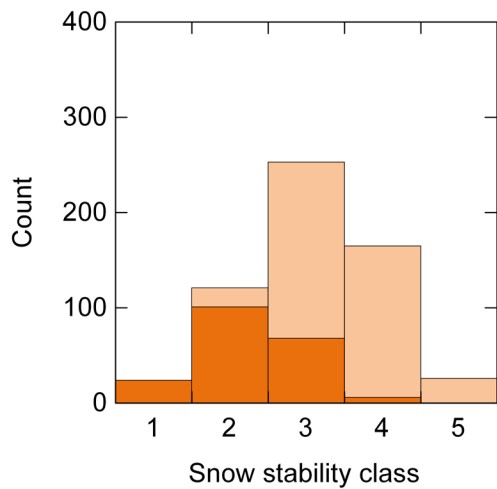

**Figure 3: Frequency of snow stability class (1: *very poor*, 2: *poor*, 3: *fair*, 4: *good*, 5: *very good*; light orange). For each class the number of cases with RB score ≤ 3, whole block release and threshold sum ≥ 5 are shown in orange (which corresponds to class A: *POOR* of the 3-level stability class scheme); *N* = 589.**

Median snow depth increased from 92 cm for the profiles rated as *very poor* to 144 cm for the profiles rated as *very good* (Figure 4) (Mann-Whitney *U*-test, $p < 0.001$). Still, there was substantial scatter, so that the correlation of stability with snow depth was rather weak ($r_{sp} = 0.24$), though statistically significant ($p < 0.001$).

Considering the 3-class stability rating scheme that combines the threshold sum, the RB score and RB release type as suggested by Schweizer et al. (2008), the three stability classes were almost equally frequent (*POOR*: 199, *FAIR*: 192, *GOOD*:
198). For the failure layers with persistent grain types, the 3-class stability was significantly lower, with *POOR* the most frequent class (41 %; 173 out of 417 cases), whereas for failure layers with non-persistent grain types the most frequent stability class was *GOOD* (54 %; $\chi^2$-test, $p < 0.001$). Likewise, with persistent grains in the failure layer *whole block* releases in the rutschblock test were observed in 172 out of 417 the cases (41 %), whereas with non-persistent grains the proportion was only 24 %, thus significantly lower (proportion test, $p < 0.001$).

Hence, both stability classifications revealed that profiles with non-persistent failure layers had better overall snow stability ratings than profiles with persistent failure layers. This finding is in line with the fact that few profiles were recorded immediately after a major snowfall, when failure layers with non-persistent grain types may actually be as weak as failure layers with persistent grain types (Reuter et al., 2019).

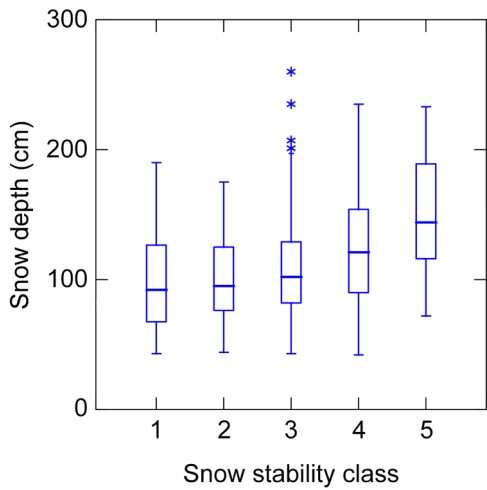

**Figure 4: Snow depth by snow stability class (1: *very poor*, 2: *poor*, 3: *fair*, 4: *good*, 5: *very good*). Boxes span the interquartile range from 1st to 3rd quartile with a horizontal line showing the median. Whiskers show the range of observed values that fall within 1.5 times the interquartile range above the 3rd and below the 1st quartile. Asterisks refer to outliers beyond the fences. $N = 589$.**

### 3.1.6  Snowpack structure index

We finally considered the snowpack structure index as introduced by Techel and Pielmeier (2014). The median value of $SNPK_{index}$ was 1.7 indicating that for about half of the profiles (52 %) the $SNPK_{index}$ class was 1 or 2 (with 1 suggesting an unfavorable snowpack structure).

With regard to profile type (Schweizer and Wiesinger, 2001), the snowpack structure index $SNPK_{index}$ was significantly higher (*U*-test; $p < 0.001$) for profiles with soft basal layers (profile types: 1 to 5) compared to profiles with profile types: 6 to 10; less than half of the profiles (46 %) had soft basal layers.

## 3.2  Danger level and stability distributions

We first present the frequency distributions for the regional forecast and the local nowcast, and the deviation between these two danger ratings. Then, we consider the 5-class stability distribution per danger level (local nowcast).

### 3.2.1  Regional forecast and local nowcast

The regional danger level as forecast in the public bulletin on the day of observation was in most cases (85 %) either *2–Moderate* or *3–Considerable* (RF; Figure 5a). This agrees with the long-term mean distribution in the region of Davos for dry-snow conditions, which is 14 %, 51 %, 34 %, 1.3 % for the forecast regional danger levels *1–Low* to *4–High*.

Observers rated the local danger level (local nowcast: LN; Figure 5a) in 22 % as *1–Low*, in 41 % as *2–Moderate*, in 35 % as *3–Considerable* and in 1.7 % as *4–High*. They agreed with the regional forecast in 70 % of the cases; in 25 % of the cases they estimated the local danger lower than forecast, in the remaining 5 % higher (Figure 5b). The agreement per danger level is shown in Appendix B.

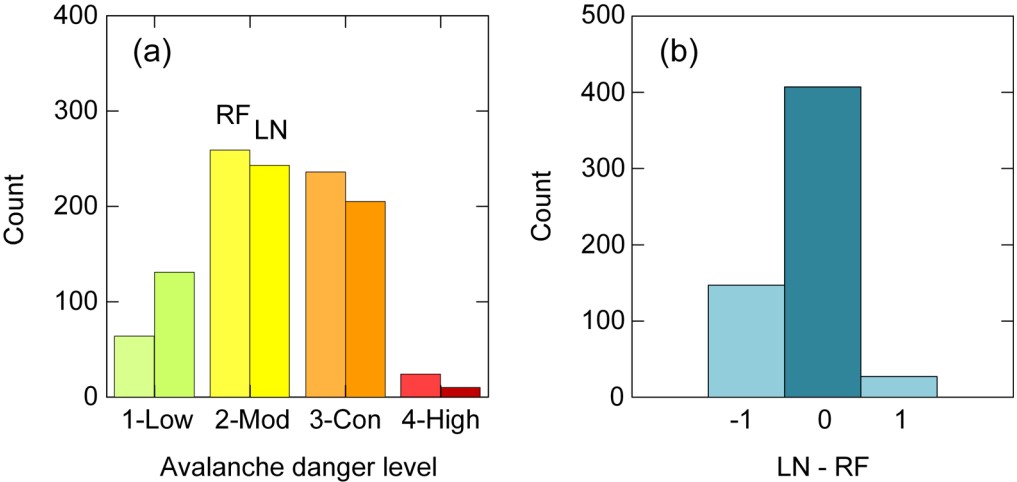

**Figure 5: (a) Frequency of avalanche danger levels (*1–Low* to *4–High*). For each danger level, on the left the regional forecast (RF; *N* = 583), on the right the local nowcast (LN, *N* = 589) is shown, for the day when the snow profile was recorded. (b) Deviation of local nowcast from regional forecast (*N* = 581; two cases with deviation of two danger levels not shown).**

### 3.2.2 Frequency of snow instability classes

Figure 6 shows the frequency distributions for the 5-class stability rating, resampled to indicate the uncertainty. The distributions of the four danger levels (LN) rated by the observers were significantly different (Mann-Whitney *U*-Test, $p < 0.001$). At danger level *1–Low*, more than 50 % of the profiles were rated as *good* or *very good*. The clear majority, almost 84 % of the profiles were rated *fair* or *good* at danger level *2–Moderate*. At the danger level *3–Considerable* about half of the profiles (49 %) were rated *poor* or *very poor*, with a very large portion of *fair* profiles. Finally, at *4–High*, the distribution is uncertain due to the low number of cases ($N = 10$) with a majority of profiles (63 %) rated as *poor* and *very poor*. Even though the stability distributions were statistically significantly different, they showed substantial overlap, in particular, for *2–Moderate* and *3–Considerable*. In both cases, the mode and median was 3, i.e. *fair*. Yet, looking at their tails on the low stability end, differences were clearly more prominent and better suited to distinguish between the different distributions.

The median sum of new snow depth in the 10 cases with *4–High* was 52 cm. In general, the sum of new snow depth (HN3d) was positively correlated with the danger level ($r_{sp} = 0.51$) and negatively with the 5-class stability rating ($r_{sp} = -0.30$).

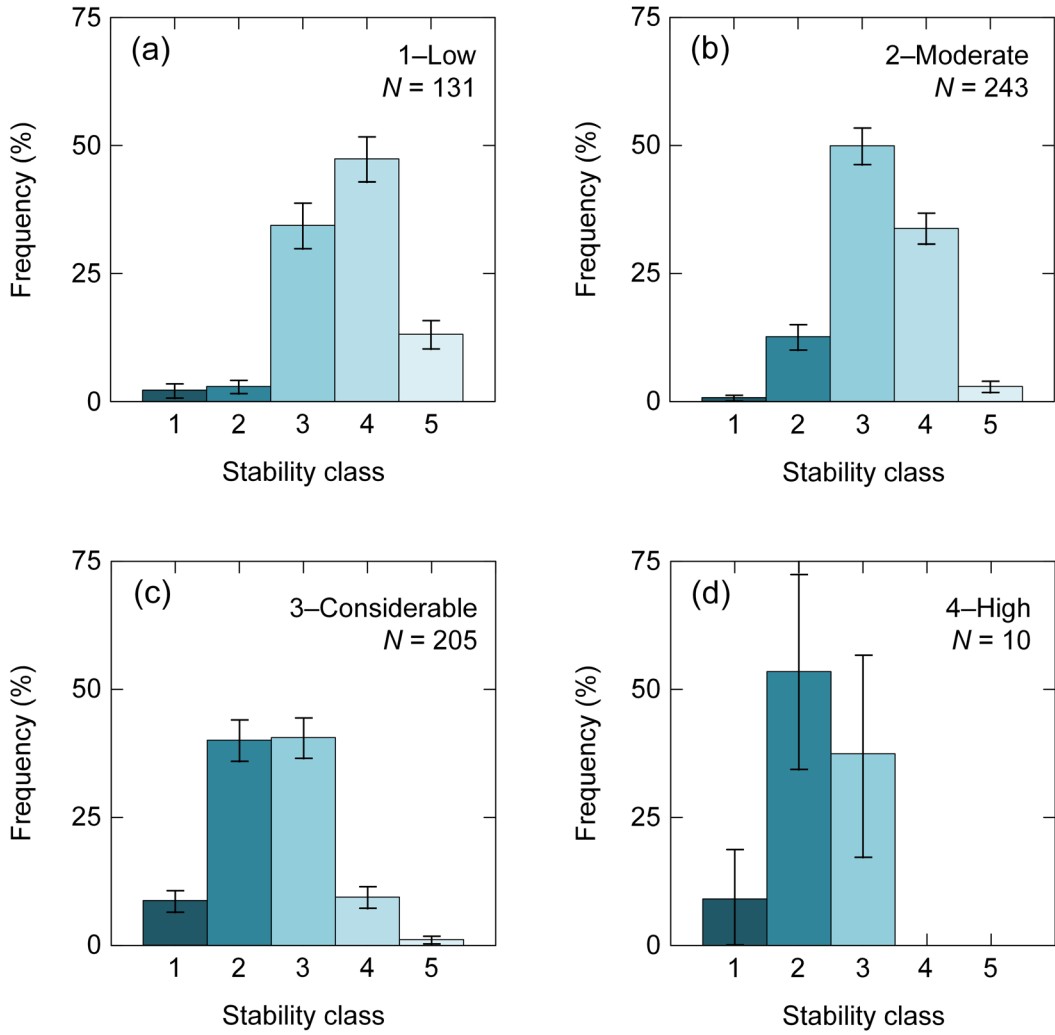

**Figure 6: Frequency distributions for the 5-class stability at the four local danger levels (a-d) from *1–Low* to *4–High* (LN, as rated by observers). Mean frequency and standard deviation (whiskers) are shown for 50 times resampling from the observed distributions.**

### 3.3    Signs of instability

In the following, we first present some general findings on the observations of signs of instability, then relate the single signs to the avalanche danger and finally, explore whether the danger levels can be characterized by signs of instability.

Either whumpfs, shooting cracks or recent avalanches were observed in 267 out of 589 cases. Whumpfs were most frequently (32 %), shooting cracks least frequently (17 %) reported. The frequency of whumpfs, shooting cracks and recent avalanching increased with increasing danger level ($r_{sp} > 0.99$; Table 3); the increases in frequency from one danger level to

the next were almost all statistically significant (proportion test, $p < 0.023$); the only two exceptions were the proportions of shooting cracks at *1–Low* vs. *2–Moderate*, and *3–Considerable* vs. *4–High*.

The strongest increase, more than 8 times, was observed for shooting cracks from *2–Moderate* to *3–Considerable*. Also, the odds ratio was 15, i.e. with shooting cracks it was 15 times more likely that the danger level was *3–Considerable* than *1–* 295 *Low* or *2–Moderate* (Table 4). For *3–Considerable* or higher the odds ratio was even larger, namely 16.

At the danger level *1–Low* signs of instability were rare; overall on 16 out of 131 days (12 %) either whumpfs, shooting cracks or recent avalanches were reported. For the danger levels *2–Moderate* to *4–High*, this proportion was 31 %, 81 % and 100 %, respectively. The seemingly inconsistent assessment of *1–Low* while signs of instability were present, were in most cases related to a low frequency of these signs and in general of triggering spots, as e.g. found in early-winter situations.

**Table 3: Frequency and relative frequency (in percent) of signs of instability per danger level (LN: local nowcast). Also given is the ratio of relative frequencies, i.e. the relative frequency of a sign of instability at a given danger level to the base rate of that sign of instability, which is given in the bottom row; total number of observations: $N = 589$.**

| Danger level | $N$ | Whumpfs | | Shooting cracks | | Recent avalanching | |
| --- | --- | --- | --- | --- | --- | --- | --- |
| | | Frequency | Ratio | Frequency | Ratio | Frequency | Ratio |
| *1–Low* | 131 | 13  9.9 % | 0.31 | 4  3.1 % | 0.18 | 4  3.1 % | 0.13 |
| *2–Moderate* | 243 | 46  19 % | 0.59 | 11  4.5 % | 0.26 | 35  14 % | 0.62 |
| *3–Considerable* | 205 | 122  60 % | 1.8 | 80  39 % | 2.3 | 88  43 % | 1.8 |
| *4–High* | 10 | 9  90 % | 2.8 | 7  70 % | 4.0 | 10  100 % | 4.3 |
| Total | 589 | 190  32 % | | 102  17 % | | 137  23 % | |

### 3.3.1 Whumpfs

Whumpfs were observed in 190 out of 589 cases, i.e. the base rate was 32 % (Table 3). In 131 out of these 190 cases (69 %) the local danger estimate was *3–Considerable* or higher; in 46 out of 190 cases it was *2–Moderate*, and in about 7 % of the cases it was *1–Low*. On the other hand, when no whumpfs were observed, the local danger estimate was in 79 % of the cases 305 *2–Moderate* or lower.

### 3.3.2 Shooting cracks

Shooting cracks were reported in 102 out of 589 cases (17 %; Table 3). When shooting cracks were recorded, observers rated the avalanche danger as *3–Considerable* or higher in 85 % of the cases; on 11 days they rated the local danger as *2–Moderate*, at four days as *1–Low*. For shooting cracks, the odds ratio increased by a factor of more than 200 from *2–Moderate* to *3–* 310 *Considerable* (Table 4); this is the largest increase and suggests that shooting cracks may have particular discrimination power.

**Table 4: Odds ratios for signs of instability and a given danger level vs. other danger levels. The odds ratio describes how much more likely a certain danger level is compared to a group of other danger levels given the observation of a sign of instability.**

| Grouping of danger levels | Odds ratio | | |
|---|---|---|---|
| | Whumpfs | Shooting cracks | Recent avalanching |
| *1–Low* vs. rest | 0.17 | 0.12 | 0.077 |
| *2–Mod* vs. *3–Con, 4–High* | 0.15 | 0.07 | 0.20 |
| *3–Con* vs. *1–Low, 2–Mod* | 7.9 | 15 | 6.4 |
| *4–High* vs. rest | 20 | 12 | - |
| *3–Con, 4–High* vs. *1–Low, 2–Mod* | 8.3 | 16 | 7.2 |

### 3.3.3   Recent avalanching

Recent dry-snow slab avalanches were reported on 137 out of 589 days (23 %) (Table 3). The local danger level estimate on those days was *1–Low* on four days, *2–Moderate* on 35 days (26 %), and *3–Considerable* or higher on the remaining 98 days (73 %). In 44 % of the days with avalanche activity, no other signs of instability were observed; this was particularly common at the lower danger levels *1–Low* (3 out 4: 75 %) and *2–Moderate* (26 out of 35: 74 %).

### 3.3.4   Whumpfs, cracks, avalanching

When all three signs of instability were observed, which was rare (in only 41 out of 589 cases: 8 %), the local danger estimate was nearly always (98 %) *3–Considerable* or higher. In cases when none of the three signs of instability were observed ($N = 322$), there were only 39 cases when the local danger level estimate was *3–Considerable*, and in most of these cases the snow stability was *fair* or lower (85 %) or HN3d > 10 cm (49 %).

On the other hand, when the local nowcast was *3–Considerable* (or higher) signs of instability were frequently observed, namely whumpfs in 60 %, shooting cracks in 39 %, and avalanches in 43 % of the cases (Table 3). For days with *2–Moderate* signs of instability were less frequent: whumpfs were triggered on 19 % of the days, shooting cracks were rare (< 5 %), and recent avalanches were observed on about every seventh day (14 %). Even though observers estimated the local danger as *1–Low*, they occasionally reported whumpfs (in 13 out of 131 cases) and on very few days even shooting cracks or recent avalanches.

### 3.3.5   Characterizing danger levels with signs of instability

For all three signs of instability, the odds ratios for *1–Low* and *2–Moderate* were < 1 (Table 4) indicating that these danger levels and the signs of instability were negatively correlated. In other words, the presence of signs of instability implied that the danger level was rather not *1–Low* or *2–Moderate*. On the other hand, the odds ratios were clearly > 1 for the danger levels

*3–Considerable* and *4–High* suggesting a positive correlation between signs of instability and these two danger levels. Even though signs of instability were more frequently associated with the danger levels *3–Considerable* and *4–High* than with *1–Low* or *2–Moderate*, assessing the danger level is not conclusive if based on signs of instability only.

Hence, in the following, we first relate the local nowcast to signs of instability with the help of a classification tree. Then we add the stability class and subsequently the 3-day sum of new snow depth as additional independent variables to ease

classification.

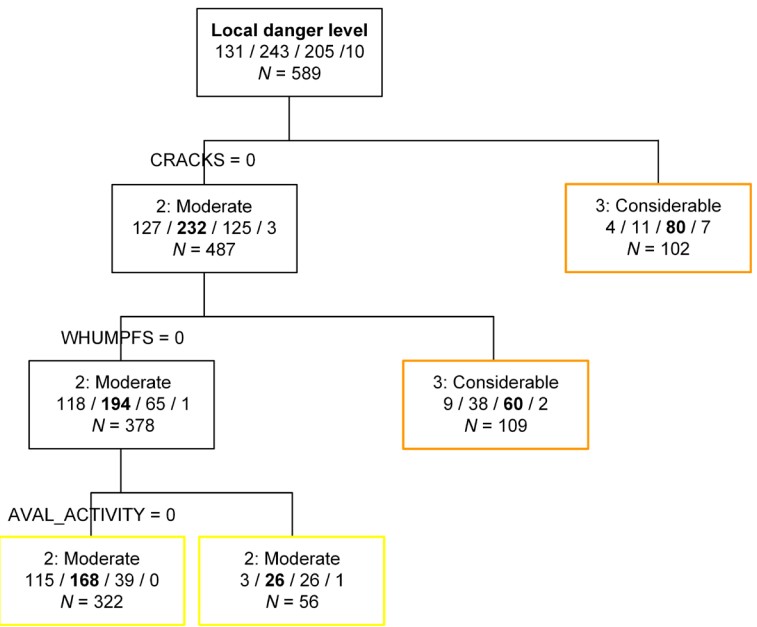

**Figure 7: Classification tree for local danger estimate (LN) based on signs of instability (whumpfs, shooting cracks and recent avalanching). The numbers indicate the frequency of the four danger levels, with the number of cases for the predicted danger level given in bold.**

The first classification tree with signs of instability only (Figure 7) showed that it was not possible to clearly discriminate between danger levels based on observed signs of instability. Based on the presence or absence of shooting cracks and whumpfs the tree suggested a classification into either *2–Moderate* and *3–Considerable* danger; *1–Low* and *4–High* could not be classified. However, in the lower left box (terminal node) of the tree in Figure 7, 115 out of 131 cases of *1–Low* were

345 found. Considering the base rate, this proportion was higher than for *2–Moderate*. The fact that *4–High* was not classified is simply due to the low number of cases, i.e. the unbalanced nature of our data set. Still, 7 out of 10 cases were found in the uppermost terminal node.

Considering the classification trees obtained with the 10 balanced resampled data sets for the danger levels *1–Low, 2–Moderate* and *3–Considerable* revealed that in 3 out of the 10 trees, the order of splits was the same as in Figure 7, namely

first cracks, then whumpfs and then avalanche activity. However, in 6 out of 10 cases the first split was the presence or absence of whumpfs. The average classification accuracy of the 10 classification trees was 56 %. In all trees there was one terminal node where the most frequent danger level was *1–Low* when no signs of instability were observed; on average 115 out 131 cases of *1–Low* were correctly classified. Also, there were always at least two terminal nodes classified as *3–Considerable*. On the other hand, in only 3 out of the 10 trees a terminal node with *2–Moderate* as most frequent end-member class existed, typically including less than 20 cases. In other words, *2–Moderate* was poorly classified, in contrast to *1–Low* and *3–Considerable*. Essentially, the trees suggested that the presence of signs of instability implies *3–Considerable*, and their absence *1–Low*. In the few cases that were correctly classified as *2–Moderate* recent avalanching was observed, but no whumpfs or cracks.

### 3.3.6 Characterizing danger levels with signs of instability and snow stability class

Using the original (unbalanced) data set and adding the 5-class stability rating as independent variable, the danger level *1–Low* could be classified (Figure 8; lower left box/end member). The danger level was *1–Low*, when there were no signs of instability and snow stability rated as *good* or *very good*, which is essentially equivalent to the absence of a critical weakness. In fact, in almost 80 % of these cases the threshold sum for the failure layer interface was ≤ 4.

Using the 10 balanced data sets instead for classification revealed that in 6 out of 10 classification trees the first split variable was the presence or absence of whumpfs, whereas the stability class only showed up once as first split variable and 4 times in the second split. Nevertheless, in all 10 classification trees all three danger levels could be classified by introducing the stability variable, a clear improvement compared to the classification trees that only included the three signs of instability. The danger level *2–Moderate* was often classified when no signs of instability were observed but the stability class was *fair* or lower, which indicates that a persistent weak layer showing intermediate stability test results existed. On the other hand, when the stability was at least *good* and no signs of instability were observed, the danger level was clearly *1–Low*. *Good* stability with either a whumpf or recent avalanching hinted toward *2–Moderate*. Clearly, cases with cracks or whumpfs plus *fair* or lower stability were classified as *3–Considerable*. The average classification accuracy of the 10 classification trees was 61 %. The best classification was obtained for *3–Considerable* (77 ± 5 %), followed by *1–Low* (58 ± 9 %) and *2–Moderate* (47 ± 14 %).

Alternatively, we also tested other stability variables, but with those not all three of the lower danger levels were classified and in general classification power in tree analysis compared to the 5-class stability variable was lower. Still, all stability related variables, the 5- and 3-class stability rating, RB score as well as the snowpack structure index were all four negatively correlated with the local danger level estimate: $r_{sp}$ = -0.50, -0.36, -0.37, -0.11; the correlation was highly significant for the two stability class variables and the RB score ($p < 0.001$).

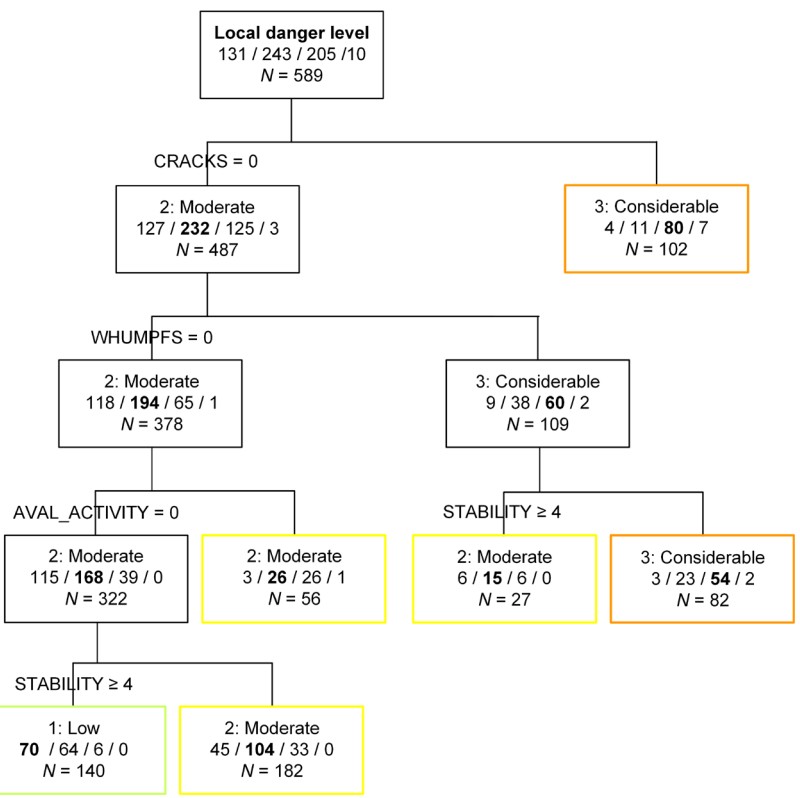

**Figure 8: Classification tree for local danger estimate (LN) based on signs of instability (whumpfs, shooting cracks and recent avalanching) and snow stability class.**

### 3.3.7 Characterizing danger levels with signs of instability, snow stability class and new snow

Finally, we added a further easily observable variable, the 3-day sum of new snow depth, as independent variable. Figure 9 shows the resulting classification tree using the original (unbalanced) data set. As with the stability class the three lower danger levels could be classified and the split rules were plausible, i.e. consistent with experience. The 3-day sum of new snow depth showed up twice as split value, once discriminating early on between days with and without new snow.

Using the 10 balanced data sets confirmed these findings. In 7 out of the 10 trees the new snow depth was the second split variable, in 3 trees it was even the first. In these 3 cases the split value was typically about 10 cm. In the other 7 trees, the first split variable was 6 times either the presence or absence of whumpfs or cracks, and once the snow stability class. Overall, the classification accuracy slightly improved, increasing to 64 %. In particular, *1–Low* was significantly better classified (77 ± 5 %), whereas the classification accuracy for the other two danger levels slightly decreased (*2–Moderate*: 44 ± 16 %; *3–Considerable*: 72 ± 5 %). Cases with danger level *1–Low* were those, as found above, with no signs of instability and *good* or higher stability, and now, in addition, no or an insignificant amount of new snow.

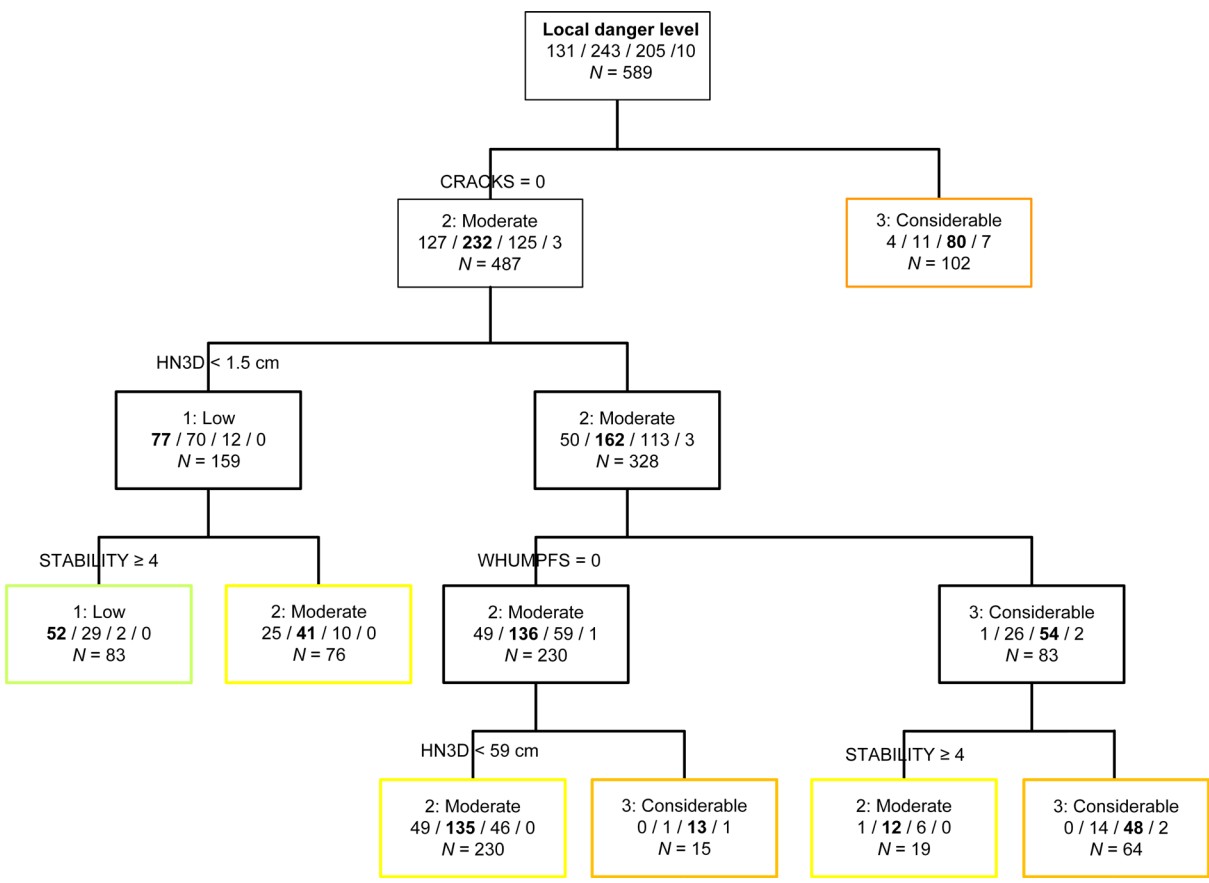

**Figure 9: Classification tree for local danger estimate (LN) based on signs of instability (whumpfs, shooting cracks and recent avalanching), snow stability class and 3d-sum of new snow depth.**

Incidentally, using the 3-day sum of new snow depth as single independent variable the median split value of the 10 trees obtained with the balanced data sets was < 11 cm. With this split value 91 % of the cases with *1–Low* were correctly classified, and 64 % of the cases with *3–Considerable*.

Finally, HN3d was 52 cm in the 10 cases with local danger level *4–High*. In general, the sum of new snow depth was positively correlated with the danger level ($r_{sp} = 0.51$) and negatively with the 5-class stability rating ($r_{sp} = -0.30$).

### 3.3.8 Characteristics of danger levels: summary

Based on the above findings we provide some typical situations that characterize the danger levels (Table 5). They are based on selected complete branches from the 30 classification trees and have to be considered as approximate description. For the danger level *4–High* we did not have enough cases for a quantitative classification analysis, but the few cases included in our

data set suggest that one can discriminate between *3–Considerable* and *4–High* based on the amount of new snow in combination with presence of signs of instability. The odds ratios given in Table 5 indicate how well the situation is related to the corresponding danger level. For the danger level *2–Moderate*, the odds ratios were relatively low, lower than for the other danger levels – in line with the findings described above.

**Table 5: Rough characterization of avalanche danger levels based on signs of instability, stability class and 3-day sum of new snow depth, with corresponding odds ratio for the corresponding danger level.**

| Danger level | Typical situation | Odds ratio |
|---|---|---|
| *1–Low* | No signs of instability | 8.7 |
| | No signs of instability and at least *good* stability | 6.4 |
| | No signs of instability and no new snow | 7.4 |
| | No signs of instability and at least *good* stability and no new snow | 9.7 |
| *2–Moderate* | No signs of instability, but 10-30 cm of new snow | 3.4 |
| | No signs of instability, but >10 cm new snow | 2.9 |
| | No signs of instability and *fair* stability | 2.7 |
| | No signs of instability, *fair* stability and new snow > 10 cm | 2.9 |
| *3–Considerable* | Shooting cracks | 10.5 |
| | Whumpfs plus *fair* or lower stability | 9.9 |
| | All three signs of instability are present. | 9.0 |
| | At least one sign of instability is present. | 11.9 |
| *(4–High)* | All three signs of instability are present and more than 35 cm of new snow. | 120 |
| | At least one sign of instability present and more than 35 cm of new snow | 89 |

## 4    Discussion

We analyzed a data set of 589 snow profiles with associated RB test and field observations that were selected from a large database of snow profiles observed in the Swiss Alps. Whereas the selection procedure may introduce potential biases, it also ensured a high quality, completeness and consistency of the data. Potential biases include, for instance, a focus on the region of Davos. However, the snowpack characteristics from the region of Davos were not different from those found in the other regions (Appendix C). In other words, the snow climate does not appear to have had a significant effect on the snowpack characteristics describing dry-snow instabilities prone to skier triggering. The limited number of profiles allowed for quality

control. This included, for each profile, to revisit the snow stability class, correcting it if inconsistently assigned, reviewing the rutschblock test results and checking failure layer depth and characteristics. Still, with 589 profiles we cover a large variety of snowpack and stability conditions from 18 winter seasons.

Almost all data represent observations that are essentially qualitative, though highly standardized. Hence, some data may be subjective and not independent. However, all observers were experienced and trained yearly according to international standards such as ICSSG (Fierz et al., 2009). In addition, during quality control any records with implausible data were discarded. On the other hand, observations of signs of instability are rather objective, but still depend on terrain travelled and the attentiveness of the field team (Jamieson et al., 2009). Finally, the local avalanche danger estimate, a key element of this study, is inherently subjective, since there is no objective measure of avalanche danger (Techel and Schweizer, 2017). Observations made during a field day will have influenced the danger assessment.

Our data do not include any information on the size of the observed avalanches and the spatial frequency of the stability information – that is essential for assessing the danger level (e.g., Techel et al., 2020). We do not know whether a single whumpf or many whumpfs were observed, or whether the weak layer found was frequently observed and known to be a problem in the area – or whether the profile was considered as not representative, i.e. was rather an outlier. Therefore, seemingly inconsistent assessments of the danger level (local nowcast) such as whumpfs at *1–Low* may reflect the lack of spatial information in our data set. For instance, several whumpfs, but a thin and discontinuous snow cover, may well justify the danger level *1–Low*, despite the observed signs of instability. In general, the local danger assessment is based on point observations; they may be insufficient for estimating the spatial frequency of instability needed for reliable danger assessment. Apart from this general scale mismatch, conditions may limit the terrain that can be travelled, e.g. at times when the avalanche situation is critical.

As we relate the local danger estimate to snowpack stability and signs of instability, there is potential for a circular argument. Given that signs of instability were recorded and at the same time the danger level estimated as *2–Moderate* and sometimes even a *1–Low* (in 12 % of the cases with *1–Low* at least one of the three signs of instability was observed), clearly indicates that observers were capable of differentiating between situations and not simply opted for *3–Considerable* as soon as they observed any sign of instability. For instance, they may have implicitly considered the spatial frequency, or the expected size of avalanches. Moreover, we are not aware of any practical method that would ensure full independence of these observational data and allow capturing characteristics of the danger levels based on observations in the field, which is the very purpose of this study.

Many snowpack characteristics we found were similar to other studies (e.g., Schweizer and Jamieson, 2003; van Herwijnen and Jamieson, 2007). For instance, the median slab depth as identified with the rutschblock test was 38 cm in our data set (Fig. 2), close to 39 cm reported by Schweizer and Jamieson (2003). Most failure layers (71 %) were ≤ 50 cm from the surface and in very few cases (1 %) the slab thickness was > 1 m. This distribution of failure layer depth or slab thickness

reflects the fact that skier stress decreases with depth and agrees with the critical range (18-94 cm) suggested in the threshold sum approach (Schweizer and Jamieson, 2007). The high percentage (71 %) of persistent grain types in failure layers confirms the prevalence of the avalanche problem *persistent weak layer* in the inner-alpine region of Davos. Even when there was a substantial amount of new snow (HN3d > 30 cm), which we argue implies the avalanche problem *new snow*, the majority of the failure layers included persistent grain types. This finding suggests that in case of snowfall the *new snow* problem is
commonly accompanied by the *persistent weak layer* problem.

The stability ratings were higher when the failure layer consisted of non-persistent grains. Also, stability increased with increasing snow depth in line with some previous studies (e.g., Schweizer et al., 2003; Zeidler and Jamieson, 2004) and supporting the approach of selecting profile locations with below average snow depth. However, Reuter et al. (2015) reported that on a particular day of extensive sampling using the snow micro-penetrometer differences in snow depth were not
necessarily related to differences in snow instability. On the other hand, they also confirmed the long-term trend across many sampling days towards higher stability with increasing snow depth.

The frequency of forecast danger levels was typical compared to the long-term statistics. The agreement of 70 % between regional forecast and local nowcast was rather low but in the range reported in previous verification studies (e.g., Techel and Schweizer, 2017). Moreover, it is not surprising that the local nowcast is often lower since it refers to a smaller
area than the regional forecast, which rather describes the highest danger level in the region.

Overall, the data set appears to be rather representative with regard to snow properties and avalanche conditions and does not indicate any particular biases – apart from potential influences of snow climate and peculiarities of the avalanche forecasting practice in Switzerland (e.g., Techel et al., 2018). Hence, in snow climates with similar types of dry-snow avalanche problems, we expect our findings to be applicable, yet corroborating them with similar analyses from other regions would
certainly be beneficial for further fostering the quantitative description of danger levels. This might require establishing international observation and reporting standards.

We then related the local observations to the local assessment of the avalanche danger level (LN). We selected the local nowcast rather than the regional forecast for two main reasons: (1) The local nowcast assessment is done after the regional forecast assessment and is no longer a forecast. Therefore, it does not include forecast errors due to, for instance, errors in the
475 weather forecast. (2) There is a scale mismatch between the local observations and the regional forecast. The regional forecast is by definition broader and cannot consider peculiarities within the region. The regional forecast has to address the highest danger prevailing in the region. Therefore, to avoid forecast errors and the scale mismatch, we related the local nowcast to the local observations.

Comparing observed signs of instability to the local danger level estimate confirmed that their occurrence increases
with increasing danger level and that they are typical at danger level *3–Considerable*. On the other hand, if no signs of instability were observed, the danger level was likely (88%) not *3–Considerable* (or higher). This finding agrees with a table

characterizing the danger levels based on typical observations that was suggested based on expert judgment by Schweizer (2003), but also with the classification tree derived by Jamieson et al. (2009) for the Columbia Mountains in Canada. Their tree predicted the danger level *3–Considerable* when whumpfs were observed – comparably to our tree in Fig. 7 – and supported by the high value of the corresponding odds ratio (Tab. 4). Adding the stability class and the 3-day sum of new snow depth improved the classification, in particular with regard to the discrimination between the lower three danger levels. Still, the danger level *2–Moderate* was not well characterized, compared to *1–Low* and *3–Considerable* that can be described by either the absence or presence of signs of instability.

We then made an attempt to extract from the classification trees some typical situations (Table 5). Again, the odds ratios illustrate that *2–Moderate* is not well defined by the simple observations we used in our analyses. At best, *2–Moderate* means that no signs of instability were observed, but there was recently some new snow or the snow stability is *fair,* i.e. there is still a persistent weak layer in the snowpack, but showing intermediate stability test results. Whereas this description is far from conclusive it indicates that in the absence of signs of instability, digging a snowpit and performing a stability test may be needed to discriminate between the lower danger levels. This interpretation is in line with the procedure suggested by, e.g., Bellaire et al. (2010).

The finding that the lower danger levels cannot easily be characterized by simple observations supports the view that only with extensive snow stability sampling the lower danger levels can be verified (Schweizer et al., 2003). Also, locations with poor stability are less frequent with lower danger levels and hence single point observations are naturally less indicative.

Whereas we did not have many stability ratings on a given day, we still attempted to derive stability distributions for the four danger levels. To this end, we assumed that on different days with the same danger level, the stability distributions are similar, so that we could combine the single stabilities from many different days into one distribution (Techel et al., 2020). This assumption is probably not fully valid as was shown by Schweizer et al. (2003) who reported slightly different stability distributions for days with the same danger level. Still, Techel et al. (2020) showed that on different days with the same danger level the stability distributions were indeed  similar, within a certain range. Nevertheless, Schweizer et al. (2003) also provided typical stability distributions for *1–Low*, *2–Moderate* and *3–Considerable*. We revisited their analysis and added their data from the winter 2002-2003 as described in Schweizer (2007). All these data are compiled in Figure 10 and allow to compare the stability distributions from this study (Fig. 6) to these previous results where up to almost 100 profiles were recorded within a few days with the same avalanche conditions and danger level in the region of Davos. Schweizer et al. (2003) also provided two stability distributions for situations that were found to be rather in-between two danger levels, which for comparison are also shown in Figure 10.

The stability distribution for *1–Low* we found, is slightly more inclined to the unstable classes compared to the distribution from the verification campaigns. There were even a few profiles rated as *very poor* and clearly more profiles rated as *fair* and less as *very good* (proportion test, $p < 0.001$). For *2–Moderate*, the trend was opposite as our distribution was

somewhat more inclined towards the stable classes: there were less profiles rated as *very poor*, and more rated as *good*

compared to the distribution from the verification campaigns (proportion test, $p < 0.017$). The distributions for *3–Considerable* were similar (*U*-test, $p = 0.25$), again with a slight trend to more stable conditions as the proportion of *very poor* profiles was lower, though the difference was not significant (proportion test, $p = 0.06$). For *4–High*, both analyses suffer from the fact that the number of cases is too low and likely include a substantial observation bias, since observers had, first of all, to consider their own safety while sampling.

The proportions of *poor* and *very poor* rated profiles clearly increased with increasing danger level. For our data set, the proportion were 5 %, 13 %, 49 % and 63 % for the danger levels *1–Low* to *4–High*, respectively. For comparison, the data from the verification campaigns revealed proportions of 2 %, 24 %, 53 % and 64 %. Hence, the main difference was the proportion at *2–Moderate* that was twice as high (proportion test, $p = 0.05$).

        Recently, Techel et al. (2020) analyzed a large data set of 4439 profiles with RB tests and established stability

distributions for the danger levels *1–Low* to *4–High*. They automatically classified stability based on the RB test results into four stability classes (very poor, poor, fair, good). Hence their stability classification approach is slightly different. Still, we can also calculate the stability classes following their approach for our data set. The resulting stability distributions using their stability classification approach are shown in the Appendix D (Figure D1). Again some differences exist, but the main features were reproduced, except for the prominent increase of the proportion of *very poor* stability test results reported by Techel et

al. (2020): 2 %, 4 %, 17 % and 38 % for the danger levels *1–Low* to *4–High*; in our data set the respective proportions were 5 %, 8 %, 20 % and 10 %. The differences exemplify that any stability distribution in part depends on the data set used and the stability scheme employed.

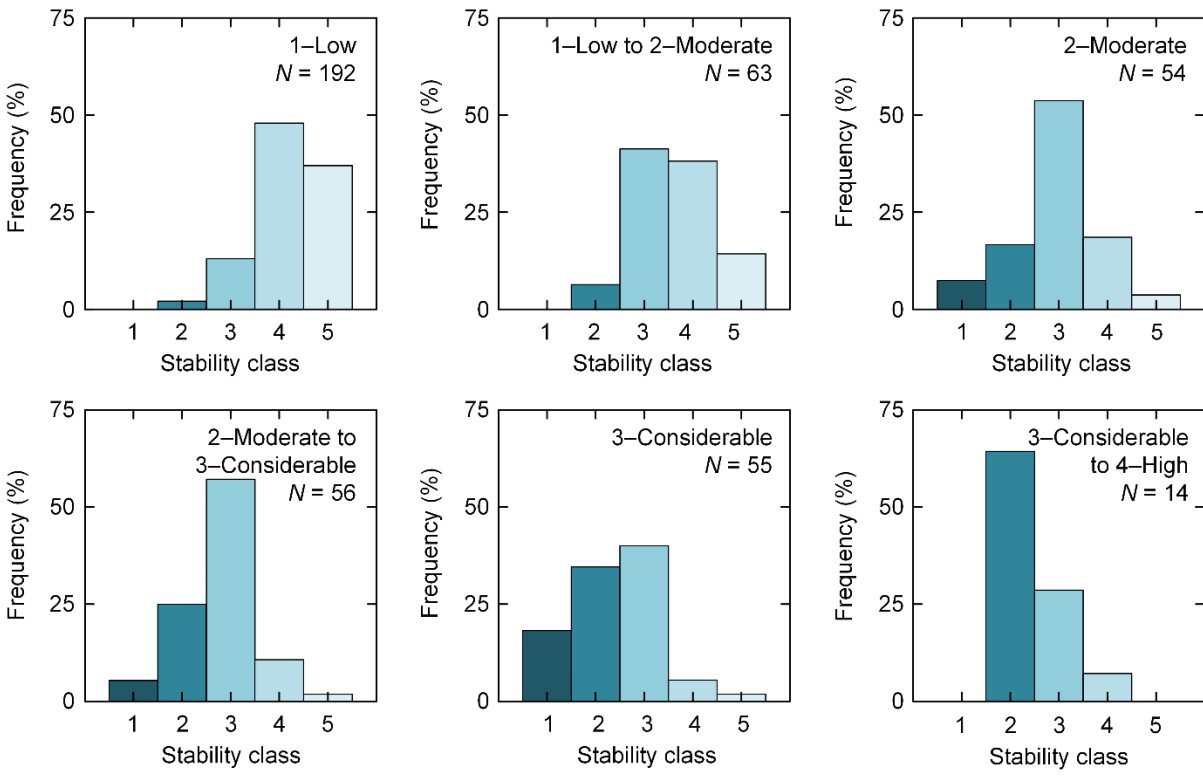

**Figure 10: Frequency distributions for the 5-class stability per danger level from the 8 verification campaigns in the winters 2001-2002 and 2002-2003; *N* = 434 (updated and adapted from Schweizer et al., 2003).**

Obviously, for the danger level *4–High* it is not feasible to derive stability distributions based on snow stability sampling by conducting snow profiles with stability tests. The increase of instability, as for instance, expressed with the proportion of *very poor* profiles is too low compared to the increases between the lower danger levels. The increase should rather be derived based on avalanche activity. For instance, Schweizer et al. (2020) reported the increase of the average number of dry-snow natural avalanches per day with danger rating *1–Low* to *4–High* as 1.3, 2.2, 4.8 and 22. This very prominent, well-known increase from *3–Considerable* to *4–High* cannot be reproduced with stability sampling. Nevertheless, it does not seem straightforward to merge the different data sets to provide a clear answer on how the frequency of locations with very poor stability increases with increasing danger level. Still, the factor is likely larger than 2, rather 3 to 5.

The ambiguities found in classification make it clear that "objective" data such as the signs of instability are not sufficient for deriving the danger level. Obviously, even after travelling a day in the field assessing the danger level includes subjectivity. This finding supports the view that any local estimate of the avalanche danger level, even by an experienced forecaster, has to be considered as an informed best guess. In fact, Techel (2020) estimated the reliability of a local nowcast to 0.88, indicating that some variation in local danger estimates exists already at the scale of a warning region (about 100 km$^2$).

A similar agreement rate between different observers was reported by Haladuick (2014). This uncertainty is inherent to "expert

verification". In other words, the lack of a quantitative approach not only illustrates that forecasting is by large empirical, it also hampers the development of numerical models and the verification of forecasts.

## 5    Conclusions

We analyzed a large data set of snow profiles including a rutschblock test (RB), concurrent observations of signs of instability and local danger estimates – to explore relations between snow instability data and avalanche danger.

With a median snow depth of 108 cm and slab thickness ranging between 25 and 52 cm (interquartile range) the snowpack exhibited characteristics related to an inner-alpine snow climate with frequent persistent weak layers – which often pose a challenge to risk assessment and forecasting. Even with recent snowfall, the majority of the failure layers included persistent grain types – suggesting that the *new snow* problem is commonly accompanied by the *persistent weak layer* problem. *Whole block* releases in rutschblock tests were frequently observed with low scores – possibly suggesting that propensity of

failure initiation and crack propagation do not decrease independently.

Snowpack observations were classified into five classes of snow stability considering RB score, RB release type and snowpack structure. Snow stability increased with increasing snow depth and was correlated with avalanche danger level (local nowcast). This local estimate of avalanche danger level agreed in 70 % with the forecast regional danger level. The snow stability frequency distributions per danger level, which we established assuming that aggregating the single observations can

represent the typical stability distribution, agreed reasonably well with the stability distributions found in previous verification studies. The proportions of *poor* and *very poor* rated profiles clearly increased with increasing danger level. For our data set, the proportion were 5 %, 13 %, 49 % and 63 % for the danger levels *1–Low* to *4–High*, respectively. These proportions agree with previous studies that also show a two- to threefold increase from one danger level to the next, at least for the lower three danger levels. However, the increase from *3–Considerable* to *4–High* cannot reliably be estimated by stability sampling and

should be assessed by the increase in natural avalanche activity, which increases by about a factor of 3 to 5.

Relating signs of instability (whumpfs, shooting cracks and recent avalanches) to the local danger estimate suggests that the presence of shooting cracks, though rarely observed, is most indicative of the danger level *3–Considerable* (or higher) as the odds ratio was highest. The presence of shooting cracks was also selected as first split in a classification tree. The most frequently observed sign of instability were whumpfs. Their presence was typical at *3–Considerable*. Still, 31 % of the

whumpfs were observed when observers rated the danger as *2–Moderate* or even *1–Low*. In the absence of signs of instability, the danger level was likely *1–Low*, provided sufficient terrain in the avalanche prone aspects and elevation band is travelled. The discrimination between the three lower danger levels *1–Low* to *3–Considerable* improved when the snow stability class and the 3-day sum of new depth were included into the tree analysis. Still, the danger level *2–Moderate* was not well

characterized. Nevertheless, we made an attempt to characterize the danger levels based on the results of the tree analysis by
providing some illustrative situations.

Whereas the frequency of signs of instability increases – as does snow instability – with increasing avalanche danger level, the characterization of danger levels with signs instability only is illustrative, but not conclusive. A quantitative description of the danger characteristics is, however, a prerequisite for applying the avalanche danger scale in forecasting and eventually providing consistent forecasts. Our findings and other recent studies suggest that a combination of data on avalanche
activity, snow instability (including its frequency distribution) and other field observations are needed to quantitively describe the wide range of snow and avalanche conditions covered by the five avalanche danger levels. In other words, for expert verification or downscaling to a local danger estimate a single set of well-defined observations is presently not available. This lack of objective verification also challenges the development of numerical forecast models.

**Data availability**

The data set of field observations of key snow profile and snow instability parameters is accessible at https://doi.org/10.16904/envidat.222 (Schweizer et al., 2021).

**Author contributions**

JS designed the study with contributions from CM, BR und FT. CM and FT extracted the data. JS curated and analyzed the data and prepared the manuscript with contributions from all co-authors.

**Competing interests**

The authors declare that they have no competing interests.

**Acknowledgements**

We are grateful for the comments by the two reviewers, Rune Engeset and Karl Birkeland, that helped us to improve the manuscript. We would like to thank all observers and SLF staff members, in particular Roland Meister, who contributed field
observations to this study.

**Financial support**

BR was supported by the Swiss National Science Foundation (grant no. P400P2_186756).

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

**Appendix A: 5-class snow stability classification**

Table A1 provides the descriptive 5-class stability classification scheme that was originally proposed by Schweizer and Wiesinger (2001). The present version does also include the rutschblock release type and the threshold sum (Schweizer and Jamieson, 2007).

**Table A1: 5-class snow stability classification scheme (adapted from Schweizer and Wiesinger, 2001).**

| Stability class | Description |
|---|---|
| **5: *very good*** | **No critical weak layers present.** <br><br>In general well consolidated (ram resistance $R$ larger than about 100 N), some soft layers (new snow or faceted crystals) near the top possible. <br><br>Faceted crystals in the lower snowpack may be present, but with $R > 100$ N (*4 Fingers* or harder). <br><br>The bottom is usually well consolidated as well, but occasionally a potentially weak base of large faceted crystals or depth hoar may exist, but is covered with a thick cohesive layer (at least 70 cm with $R > 200$ N). <br><br>Profile type: 4, 6 and 10 <br><br>**Rutschblock score: 6 or 7 (independent of release type); 0-2 Lemons** |
| **4: *good*** | Weak layers may be present, but not very prominent, e.g. showing no clean shears. <br><br>In general, well consolidated middle part with $R > 100$ N, or prominent hard crust of a few centimetres thickness in the upper third of the snowpack. <br><br>At the bottom, a potentially weak base with large faceted crystals or depth hoar may exist, but is covered with cohesive snow (at least 50 cm with $R > 100$ N). <br><br>The snowpack may still fail if applying high loads on weaknesses (not prominent), or on top of the depth hoar base. <br><br>Profile type: 2, 3, 4, and 6; <br><br>Rutschblock score: 5 (partial release) and 0-4 Lemons; or 6 (independent of release type) and 3 Lemons |
| **3: *fair*** | Weak layers are present, showing clean shears, but transitional RB scores (4,5). Weak layers often consist of rounded persistent forms. <br><br>Some soft layers with $R \approx 40$ N present (except new snow on top), but most of the snowpack is fairly well consolidated. <br><br>Profile type: 2, 3, 4, 8, 9; occ. 7 <br><br>Rutschblock score: 4 (whole block); or 5 (whole block) and 4-5 Lemons; or 3 (partial release). |

| 2: *poor* | Prominent weak layers present, showing clean shears. Weak layers of surface hoar or faceted crystals, larger than 1 mm, or failures within the new or partly settled snow, or new snow on crust. |
|---|---|
| | Hardness of slab is $R < 40$ N (*Fist* to *4 Fingers*). |
| | Some well consolidated parts may exist ($R = 100$–$300$ N), but the thickness of these layers is less than 30 cm. |
| | Profile type: 1, 2, 5, 7, 8 and 9 |
| | Rutschblock score: 2 or 3 (whole block) |
| 1: *very poor* | **Prominent weak layers present.** |
| | Weak layers of surface hoar or faceted crystals, larger than 1-2 mm sandwiched between harder layers, or facets on crusts. |
| | The bottom is frequently weak, occasionally covered with only one cohesive slab layer. The ram resistance may be low from top to bottom ($R \approx 20$ N). |
| | In general, ram resistance above the weak layer is $R < 50$ N, often *Fist*. |
| | There are no hard layers with $R > 150$ N present, crusts are usually thin and do not show up in the ram profile. |
| | Profile type: 1, 5, 7 and 9 |
| | **Rutschblock score: 1 or 2 (whole block)** |


## Appendix B: Differences between regional forecast and local nowcast

In addition to Figure 5b, we show in Figure B1 the agreement and deviations for each of the danger levels *1–Low* to *4–High* between regional forecast and local nowcast.

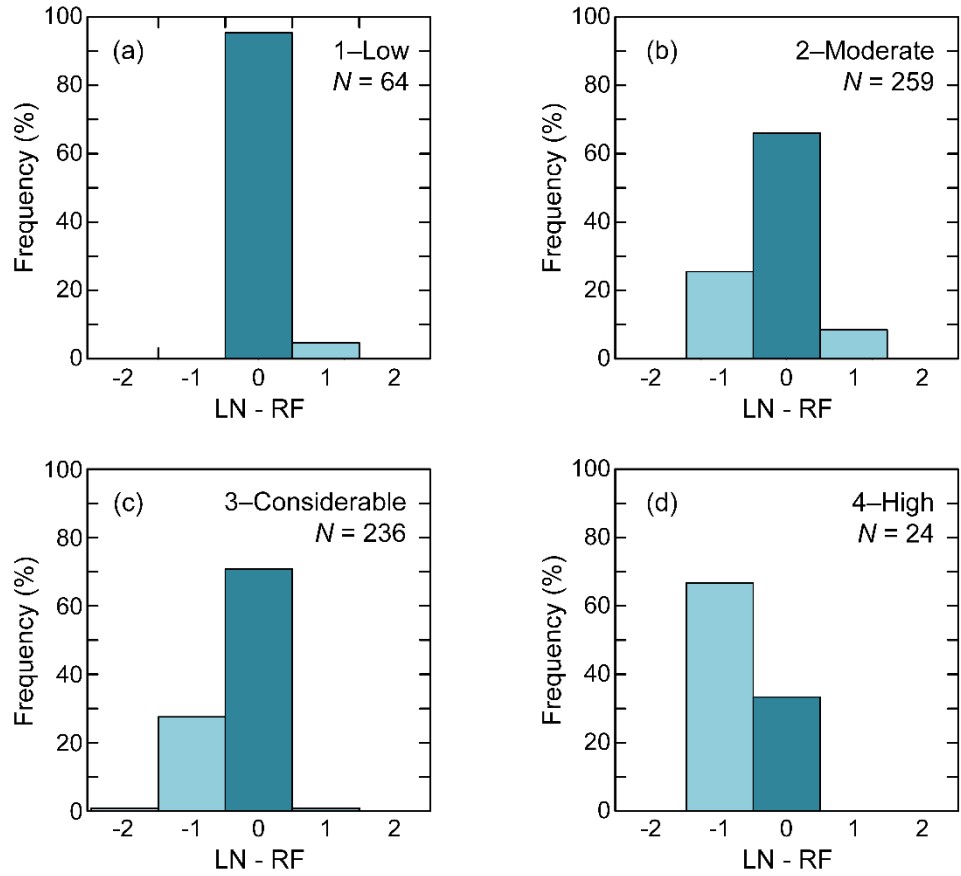

**Figure B1: Agreement and deviation between regional forecast and local nowcast for each of the four danger levels (a-d) from *1– Low* to *4–High* (*N* = 583). A deviation of -1 indicates that the danger level of the local nowcast was one level lower than the regional forecast, and vice versa for a deviation +1.**

## Appendix C: Regional differences in snowpack characteristics

From 589 field observations, 561 (95 %) were from the region of Davos. To check whether the remaining 5 % that were not recorded in the region of Davos introduced any bias, we contrasted key snowpack properties from the two groups (Figure C1, Table C1). The profile characteristics were very similar. In particular snow depth, a good indicator of snow climate, was not

significantly different (median snow depth 120 cm vs. 108 cm; $p = 0.13$). Also, the snowpack structure index, which relates to the importance of faceting (or the presence of persistent weak layers) is not different either (1.78 vs. 1.70; $p = 0.7$).

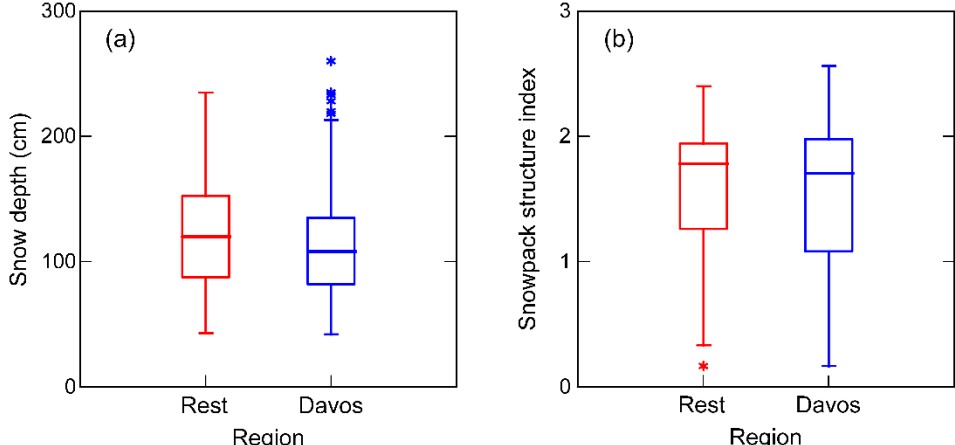

**Figure C1: Comparison of selected profile characteristics from the region of Davos ($N = 561$) and from other regions (Rest, $N = 28$): (a) Snow depth and (b) snowpack structure index. For median values and $p$-value see Table B1.**

**Table C1: Comparison of selected profile characteristics from the region of Davos ($N = 561$) and the other regions ($N = 28$). Median is shown (mode in the case of FL grain type) and level of significance based on non-parametric Mann-Whitney U-test.**

| Parameter | Rest | Davos | $p$ |
|---|---|---|---|
| Elevation (m a.s.l.) | 2488 | 2470 | 0.367 |
| Slope angle (°) | 34 | 33 | 0.301 |
| Snow depth (cm) | 120 | 108 | 0.131 |
| Slab thickness (cm) | 41 | 38 | 0.052 |
| FL grain size, avg (mm) | 1 | 1 | 0.643 |
| FL grain size, max (mm) | 1.5 | 1.5 | 0.846 |
| FL grain type | facets | facets | 0.872 |
| RB score | 4-5 | 4 | 0.737 |
| 5-class stability | fair | fair | 0.761 |
| SNPK$_{index}$ | 1.78 | 1.70 | 0.702 |

**Appendix D: 4-class stability classification per danger level**

For comparison, the distributions for the 4-class stability per danger level is shown (Figure D1). The stability classification into four classes (very poor, poor, fair, good) was introduced by Techel et al. (2020) and is solely based on RB score and RB release type, which allows for automated classification.

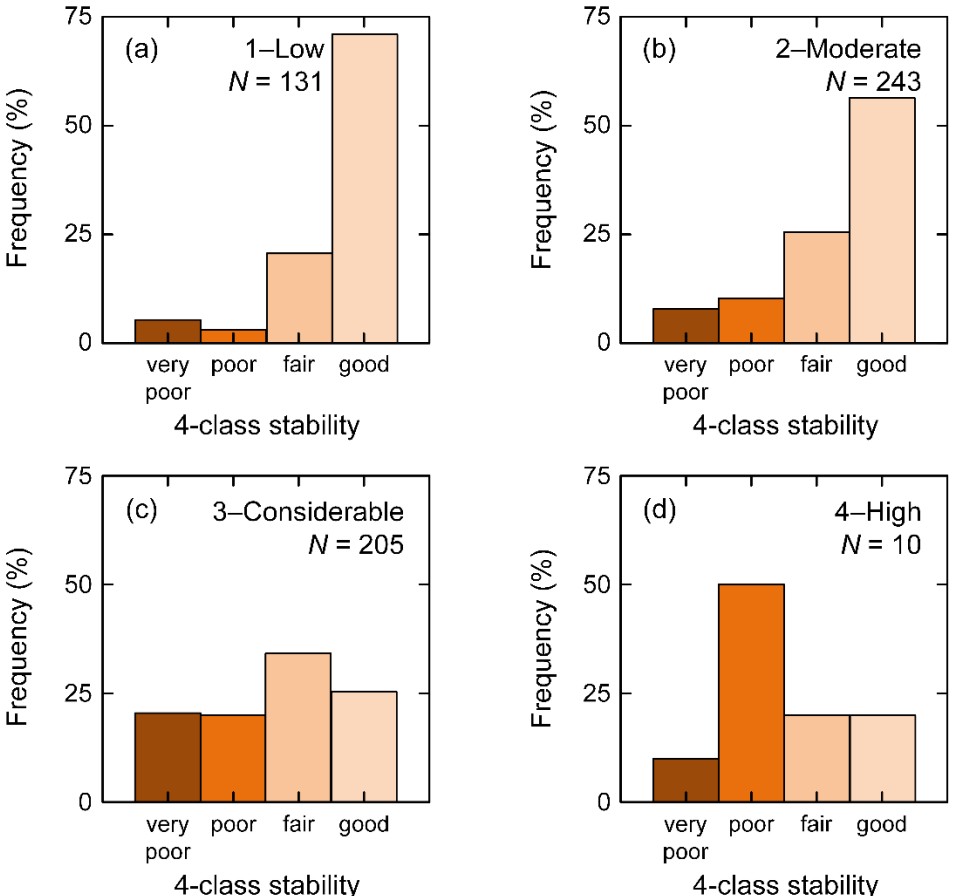

**Figure D1: Frequency distributions for the 4-class stability rating (Techel et al., 2020) at the four local danger levels (a-d) from *1–Low* to *4–High* (LN, as rated by observers); *N* = 589.**