# Peer review of "Avalanche danger level characteristics from field observations of snow instability"

_The Cryosphere, 2020_

## Referee Comment (RC1) · Rune Engeset (Referee) · 25 Jan 2021

Review of the manuscript Avalanche danger level characteristics from field observations of snow instability by Jürg Schweizer, Christoph Mitterer, Benjamin Reuter, and Frank Techel.

https://tc.copernicus.org/preprints/tc-2020-350/

**General comments**

The paper is well written, has a clear scope and analysis, and presents new results of scientific and practical value. It is an important contribution towards improving avalanche danger assessments and forecasting. It helps scientists and practitioners to assess and define avalanche danger, with practical implications ranging from developing machine learning tools to crowd-sourcing and interpreting signs and observations in the field. It also sheds light on the timing and combination of avalanche problems (new snow and persistent weak layer) based on observations, and provides directions for further research. The language, figures and tables are of high quality and the flow of the manuscript is sound and logical.

I recommend the editors to publish the paper after minor revisions.

**Specific comments**

Here are specific comments, which could be used to improve the manuscript:

- Not only stability, but also distribution/frequency of instability *and* the avalanche size affect the LN data and thus the analysis. The effect of not accounting for this on the results could be described further in the discussion and mentioned in the conclusions, if possible
- The data comes from a limited geographical area, and the study would ideally include data from other snow climates and regions of the world. I guess such data are difficult to come by, but it could be a recommendation to include more data in future studies (e.g. explain to other scientist how to collect and share data in a common data set for an analysis of more a regional or even continental/global perspective). It would also be interesting to know which other observations the authors would recommend for future studies, to improve the observational basis for quantifying avalanche danger (properties of the weak layer or the slab?)
- The introduction could explain how this study adds to and differs from the studies by Techel et al. (2020) and Schweizer et al. (2020)
- In chapter 3.1.2, could you include statics for quality of fracture plane (smooth, rough, irregular)?
- I would suggest reversing the x-axis in the figure in the appendix (have very poor to the left and good to the right) to align with the other figures in the manuscript, and to insert this figure (as well as its results) in the results chapter. It could also be beneficial to add a paragraph discussing, comparing or summarising the results of the three different scales used for stability (categorising stability into 3, 4 and 5 classes)

**Technical corrections**

Now follows specific comments, with reference to line numbers in the manuscript:

- #30-37 This is a description of a regional forecast, I suggest you add the word "regional" somewhere, and briefly describes how regional warnings are different from other forecasts (object-based / slope-scale) and, if possible, if this study may be valuable also to slope-scale assessments/forecasts
- #32 I could not easily find the EAWS 2019 in the reference list (it is on #570)

- #39 Could you improve the English of the sentence starting with "Even"?
- #59-60 Consider rewrite and simplify the first sentence, e.g. "but it was not possible to distinguish 2–Moderate from 1–Low"
- #84 Explain why you selected dry snow conditions only (dry slab selection could also be explained in #113)
- #104 Could you improve the definition of the "adjacent layer" and explain when it is above and below the failure interface?
- #108 Explain why you derived three stability classes her, while using five classes above (#95) and even 4 classes in the appendix
- #171-174 Please simplify this description
- #242 Please explain the acronym "RF" (presumably regional forecast)
- #245 Please clarify: Did the observer estimate the LN for the same region / area as the RF or is LN valid for a smaller area than RF and thus probably biased towards lower danger levels (presuming that RF is the highest danger level in the region)?
- #277 I presume "even" could be deleted (or is the sentence incomplete?)
- #367 Could you improve the sentence "the split variables and values were plausible"?
- #436 Could you improve the sentence "only in 1 out 5 cases differences in snow depth were indicative of snow instability"?
- #523 Please explain how/why this poses a challenge to forecasting?

Rune Engeset, 24 January 2021

---

## Referee Comment (RC2) · Karl W. Birkland (Referee) · 27 Feb 2021

I selected "Major Revisions" for this paper simply because some of my comments suggest that the paper may benefit from re-analyzing some of the data. However, I do not think that such a re-analysis should be overly difficult, and so my review probably falls somewhere between "major revisions" and "minor revisions".

This research aims "to characterize the avalanche danger levels based on expert field observations of snow instability." This is an important goal and is something that would be beneficial for avalanche forecasters and for a better understanding avalanche danger. The authors utilize an interesting, unique, and high-quality data set. While I believe the authors have produced an interesting paper, I believe it would benefit from some

changes and clarifications. I have four major comments:

1. First, when I was initially and quickly reading the paper and looking at the title, I assumed that the "avalanche danger" referred to in the title was the regional avalanche danger rating. However, this is not the case. Rather the "avalanche danger rating" is really the local nowcast provided by the observer. As pointed out in the paper, there is a scale mismatch between a local rating for a particular area and a regional avalanche danger rating. In addition, the authors point out that they are utilizing a local nowcast in their abstract. Despite this, I believe the authors should more clearly define the differences between these two ratings and if they decide to utilize the local nowcast avalanche danger then that should be specified in the title.

2. Second, along the lines of my first comment, the authors acknowledge that there is circular reasoning in their data since the observers are making snowpack and avalanche observations and are also assigning the local avalanche danger. Undoubtedly the observers are taking their observations into account when they are assigning the local avalanche danger. I believe this is somewhat problematic. In this scenario the authors may actually be testing what snowpack observations the observers happen to associate with a particular local avalanche danger rating rather than the more general question of which snowpack observations are associated with a danger level. I am wondering why the authors didn't simply compare the local observations to the regional avalanche danger assigned for that region for that day? That way there would be independence between the snowpack observer and the assessment of the avalanche danger, and the results would reflect the frequency of making these observations for a given regional danger level rating. I would suggest either utilizing the regional danger ratings or providing a solid rationale for not using them in the paper.

3. Third, this paper utilizes a unique dataset of snow profiles and observations from Switzerland. Approximately 95% of the data are from the region of Davos. Later the authors explain that some of their results, such as the predominance of persistent weak layers, may be because so much of their data are from the Davos area. I believe the

paper would benefit from using only those snow profiles and observations from around Davos, rather than a highly unbalanced dataset consisting of almost all profiles from Davos and then 5% from other areas. This would still retain 95% of the data but would remove some of the variability introduced by the other 5% of the data.

4. My fourth comment relates back to my third. The authors spend considerable effort (and content in their paper) characterizing the snow profiles. This provides some interesting results that I believe should be retained, but it is outside the primary stated goal of the paper. As stated above, I believe this analysis would have more meaning if the data were restricted to just the 95% of data from the region of Davos. Then the snowpack characterization part of the paper can provide a characterization of the Davos area snowpack rather than "mostly" the Davos area snowpack with 5% of the profiles and observations from other areas. In addition, this part of the paper should be better highlighted in the abstract and perhaps the title as well since nearly as much attention is paid to this snowpack characterization as is paid to the relationship of the snow profiles and observations to the danger levels. I think that this "characterization of a snowpack" in a region is quite valuable and will set a baseline for future work which could compare this characterization against the characterization of the snowpack in other regions or other countries.

While I do have some substantial comments that I believe the authors should address, I do think that this is important work and that it should be published following revisions.

In addition to the above comments, I have some more minor comments and suggested typographical corrections:

Line 29: Delete "at times"

Line 40: Delete "were" and replace with "have been"

Line 68: Replace "inexistent" with "nonexistent"

Line 84: Can you provide more specifics about how you defined "an experienced ob-

server"?

Line 97: After reading this paragraph, I am still not certain how the failure layer and adjacent layers are defined. I understand how the authors come up with the failure interface, but how do they necessarily define the failure layer? Was this done manually by the authors? And, with the adjacent layer, was this typically just the layer adjacent to the failure interface that was not the failure layer? I think this might be less confusing if the authors talked about the failure interface, and then the layer above that interface and the layer below that interface. They could then also quantify how often the "failure layer" is below or above the layer interface.

Line 115 – It seems like assigning 3+ to Considerable is arbitrary. Why not apply High here since 2+ is also assigned to Considerable?

Line 199-200 and line 525 – I don't think that a decrease in the whole block release of RBs with increasing RB number is necessarily related to a decrease in crack propagation propensity. This may have more to do with the increased damage to the slab caused by harder jumping on the RB that causes partial block releases. Given the complexities involved, I don't think the authors can draw such a definitive conclusion from their data. While I don't have a good dataset to either confirm or refute this conclusion, I have anecdotally seen seasons where PSTs consistently propagate to end – indicating the propensity for crack propagation – for a long time after other tests have indicated that failure initiation is far less likely.

Line 528: I assume the "avalanche danger level" in the sentence on this line is the local avalanche danger? As stated in one of my primary comments, it would be helpful to make sure to careful differentiate between the local avalanche danger rating and the "avalanche danger rating". I always assume this latter term is associated with the regional avalanche danger rating.

Line 529: Along the lines of my comment above, the authors state that the local avalanche danger rating agrees with 70% of the regional danger ratings. It would

be interesting to more thoroughly compare how those two ratings at the two different scales differ.

Line 454: Delete "and" and replace with "or"

Karl Birkeland

---

## Author Comment (AC1) · 4 May 2021

**Reply to Reviewer #1**

We thank the reviewer, Dr. Engeset, for his insightful comments. Below we provide answers to the major points and indicate how we will improve the manuscript.

*General comments*
*The paper is well written, has a clear scope and analysis, and presents new results of scientific and practical value. It is an important contribution towards improving avalanche danger assessments and forecasting. It helps scientists and practitioners to assess and define avalanche danger, with practical implications ranging from developing machine learning tools to crowd-sourcing and interpreting signs and observations in the field. It also sheds light on the timing and combination of avalanche problems (new snow and persistent weak layer) based on observations, and provides directions for further research. The language, figures and tables are of high quality and the flow of the manuscript is sound and logical.*
*I recommend the editors to publish the paper after minor revisions.*

*Specific comments*
*Here are specific comments, which could be used to improve the manuscript:*
*- Not only stability, but also distribution/frequency of instability and the avalanche size affect the LN data and thus the analysis. The effect of not accounting for this on the results could be described further in the discussion and mentioned in the conclusions, if possible.*

We agree and will include these points into the Discussion section of the revised manuscript.

*- The data comes from a limited geographical area, and the study would ideally include data from other snow climates and regions of the world. I guess such data are difficult to come by, but it could be a recommendation to include more data in future studies (e.g. explain to other scientist how to collect and share data in a common data set for an analysis of more a regional or even continental/global perspective). It would also be interesting to know which other observations the authors would recommend for future studies, to improve the observational basis for quantifying avalanche danger (properties of the weak layer or the slab?)*

We agree that our data comes from a limited geographical area. As described in the Discussion section there is considerable agreement with regard to snowpack properties with previous studies, for instance, by Schweizer and Jamieson (2003). Their study included profiles from the Columbia Mountains of western Canada and Switzerland. This agreement suggests that characteristics of instability in a snow profile may not depend so much on snow climate – on the other hand, of course, their frequency of occurrence will depend on climate.
With regard to data collection, we think the issue is not so much about the type of data to be collected, but consistency. Hence most important are proper observations standards and observer training. Of course, the EAWS may play role here in establishing standards such as the SWAG published by the American Avalanche Association.

*- The introduction could explain how this study adds to – and differs from – the*

The present study includes the profile characteristics as well as the relation of the danger level to signs of instability. It even aims at describing the danger level based on observations, which is not possible based on the study by Techel et al. (2020) where the frequency distribution of instabilities needs to be known. The study by Schweizer et al. (2020) solely focussed on avalanche size.

*-          In chapter 3.1.2, could you include statics for quality of fracture plane (smooth, rough, irregular)?*

The quality of fracture plane is known to be less indicative of stability than, for instance, the RB release type (e.g., Schweizer et al., 2008). Overall, 64 % of the tests showed a smooth fracture plane, 21 % a rough one, and 15 % of the failures were irregular. For rutschblock score 1 to 3, the proportion of smooth fracture planes was slightly higher than average, and vice versa for RB scores 4 to 6 (Figure R1). On the other hand, the proportion of irregular fracture planes clearly increased with increasing RB score. In conclusion, the observation of an irregular fracture plane indicates rather favourable conditions, whereas any other type of fracture plane does not allow any firm conclusions on stability.
We will add this information in the revised manuscript.

[Figure]

Figure R1: Proportions of the three fracture plane types per rutschblock score ($N = 581$).

*-          I would suggest reversing the x-axis in the figure in the appendix (have very poor to the left and good to the right) to align with the other figures in the manuscript, and to insert this figure (as well as its results) in the results chapter. It could also be beneficial to add a paragraph discussing, comparing or summarising the results of the three different scales used for stability (categorising stability into 3, 4 and 5 classes)*

We will change the x-axis as suggested and consider its inclusion in the main text. Our original idea was to provide the figure in the Appendix primarily for illustration and to facilitate the

comparison to the previous study by Techel et al. (2020). Including it in the main text has the potential for confusion and would require a more thorough introduction and more background information on this type of stability classification.

*Technical corrections*
*Now follow specific comments, with reference to line numbers in the manuscript:*
*-        #30-37 This is a description of a regional forecast, I suggest you add the word "regional" somewhere, and briefly describes how regional warnings are different from other forecasts (object-based / slope-scale) and, if possible, if this study may be valuable also to slope-scale assessments/forecasts.*

We will explicitly refer to the regional scale in the revised manuscript. We implicitly do so in line 35. Stability evaluation is valuable for slope-scale assessments, but as pointed out in line 36 the slope scale or object-based assessments are different from regional forecasts.

*-        #32 I could not easily find the EAWS 2019 in the reference list (it is on #570)*

Thanks for pointing this out. We will remove this inconsistency in the revised manuscript.

*-        #39 Could you improve the English of the sentence starting with "Even"?*

We will revise the sentence.

*-        #59-60 Consider rewrite and simplify the first sentence, e.g. "but it was not possible to distinguish 2–Moderate from 1–Low"*

We will revise this sentence.

*-        #84 Explain why you selected dry snow conditions only (dry slab selection could also be explained in #113)*

Stability evaluation for dry-snow conditions is of particular relevance to mitigate the persistent weak layer avalanche problem type. Profile data are primarily indicative of dry-snow conditions. Also, signs of instability such as whumpfs and shooting cracks relate to dry-snow slab avalanches. We will provide somewhat more rationale in the revised manuscript.

*-        #104 Could you improve the definition of the "adjacent layer" and explain when it is above and below the failure interface?*

We follow the approach that was introduced by Schweizer and Jamieson (2003). In all profiles, the failure interface was reported. While in most cases the weak layer is obvious, we considered the softer of the two layers as the failure layer (FL) and the layer across the failure

interface as the adjacent layer. If there was no difference in hardness, we selected the lower layer as the failure layer, and the layer above the failure interface as the adjacent layer.

- *#108 Explain why you derived three stability classes her, while using five classes above (#95)and even 4 classes in the appendix*

Traditionally, five classes were used as described by Schweizer and Wiesinger (2001). In various subsequent studies simplifications were sought, often to ease interpretation or to make it amenable for automatization. As with any classification the choice of classes is partly arbitrary. We will provide some more details in the revised manuscript.

- *#171-174 Please simplify this description*

We will revise this description.

- *#242 Please explain the acronym "RF" (presumably regional forecast)*

We will introduce RF in the Data and Methods section of the revised manuscript.

- *#245 Please clarify: Did the observer estimate the LN for the same region / area as the RF or is LN valid for a smaller area than RF and thus probably biased towards lower danger levels (presuming that RF is the highest danger level in the region)?*

Many thanks for pointing out this scale issue, which we will discuss it in the revised manuscript. We also provide a figure in the Reply to Reviewer #2 that illustrates this issue (Figure R2).

- *#277 I presume "even" could be deleted (or is the sentence incomplete?)*

We will revise this sentence.

- *#367 Could you improve the sentence "the split variables and values were plausible"?*

We will improve the sentence.

- *#436 Could you improve the sentence "only in 1 out 5 cases differences in snow depth were indicative of snow instability"?*

We will clarify this sentence and provide more detail on the study by Reuter et al. (2015).

- *#523 Please explain how/why this poses a challenge to forecasting?*

Persistent weak layers can linger for a long time in the snowpack and represent the failure layer in skier-triggered avalanches. However, it is challenging to communicate a danger (or the avalanche problem type: persistent weak layer) that is "invisible" and often only sporadically present.

**References**

Reuter, B., van Herwijnen, A., Veitinger, J., and Schweizer, J.: Relating simple drivers to snow instability, Cold Reg. Sci. Technol., 120, 168-178, https://doi.org/10.1016/j.coldregions.2015.06.016, 2015.

Schweizer, J., and Wiesinger, T.: Snow profile interpretation for stability evaluation, Cold Reg. Sci. Technol., 33, 179-188, https://doi.org/10.1016/S0165-232X(01)00036-2, 2001.

Schweizer, J., and Jamieson, J. B.: Snowpack properties for snow profile analysis, Cold Reg. Sci. Technol., 37, 233-241, https://doi.org/10.1016/S0165-232X(03)00067-3, 2003.

Schweizer, J., McCammon, I., and Jamieson, J. B.: Snowpack observations and fracture concepts for skier-triggering of dry-snow slab avalanches, Cold Reg. Sci. Technol., 51, 112-121, https://doi.org/10.1016/j.coldregions.2007.04.019, 2008.

Schweizer, J., Mitterer, C., Techel, F., Stoffel, A., and Reuter, B.: On the relation between avalanche occurrence and avalanche danger level, Cryosphere, 14, 737-750, https://doi.org/10.5194/tc-14-737-2020, 2020.

Techel, F., Müller, K., and Schweizer, J.: On the importance of snowpack stability, the frequency distribution of snowpack stability, and avalanche size in assessing the avalanche danger level, Cryosphere, 14, 3503-3521, https://doi.org/10.5194/tc-14-3503-2020, 2020.

---

## Author Response (AR1)

**Reply to Reviewer #1**

We thank the reviewer, Dr. Engeset, for his insightful comments. Below we provide answers to the major points and indicate how we improved the manuscript.

*General comments*
*The paper is well written, has a clear scope and analysis, and presents new results of scientific and practical value. It is an important contribution towards improving avalanche danger assessments and forecasting. It helps scientists and practitioners to assess and define avalanche danger, with practical implications ranging from developing machine learning tools to crowd-sourcing and interpreting signs and observations in the field. It also sheds light on the timing and combination of avalanche problems (new snow and persistent weak layer) based on observations, and provides directions for further research.The language, figures and tables are of high quality and the flow of the manuscript is sound and logical.*
*I recommend the editors to publish the paper after minor revisions.*

*Specific comments*
*Here are specific comments, which could be used to improve the manuscript:*
*-        Not only stability, but also distribution/frequency of instability and the avalanche size affect the LN data and thus the analysis. The effect of not accounting for this on the results could be described further in the discussion and mentioned in the conclusions, if possible.*

We agree and now address the issue that we lack information on avalanche size and spatial frequency of instability in the Discussion section (line 426).

*-        The data comes from a limited geographical area, and the study would ideally include data from other snow climates and regions of the world. I guess such data are difficult to come by, but it could be a recommendation to include more data in future studies (e.g. explain to other scientist how to collect and share data in a common data set for an analysis of more a regional or even continental/global perspective). It would also be interesting to know which other observations theauthors would recommend for future studies, to improve the observational basis for quantifying avalanche danger (properties of the weak layer or the slab?)*

We agree that our data come from a limited geographical area. As described in the Discussion section there is considerable agreement with regard to snowpack properties with previous studies, for instance, by Schweizer and Jamieson (2003). Their study included profiles from the Columbia Mountains of western Canada and Switzerland. This agreement suggests that characteristics of instability in a snow profile may not depend so much on snow climate – on the other hand, of course, their frequency of occurrence will depend on climate. We now provide additional information in Appendix B.

With regard to data collection, we think the issue is not so much about the type of data to be collected, but consistency. Hence most important are proper observations standards and observer training. Of course, the EAWS may play role here in establishing standards such as the SWAG published by the American Avalanche Association. We added a statement calling for consistency and observation standards (lines 468-469).

- *The introduction could explain how this study adds to – and differs from – the studies by Techel et al. (2020) and Schweizer et al. (2020)*

The present study includes the profile characteristics as well as the relation of the danger level to signs of instability. Moreover, it aims at describing the danger level based on observations, which is not possible based on the study by Techel et al. (2020) where the frequency distribution of instabilities needs to be known. The study by Schweizer et al. (2020) solely focussed on avalanche size. The two recent studies are described in the Introduction section (lines 62-68).

- *In chapter 3.1.2, could you include statics for quality of fracture plane (smooth, rough, irregular)?*

The quality of fracture plane is known to be less indicative of stability than, for instance, the RB release type (e.g., Schweizer et al., 2008). Overall, 64 % of the tests showed a smooth fracture plane, 21 % a rough one, and 15 % of the failures were irregular. For rutschblock scores 1 to 3, the proportion of smooth fracture planes was slightly larger than average, and vice versa for RB scores 4 to 6 (Figure R1). On the other hand, the proportion of irregular fracture planes clearly increased with increasing RB score. In conclusion, the observation of an irregular fracture plane indicates rather favourable conditions, whereas any other type of fracture plane does not allow any firm conclusion on stability.
We added this information in the revised manuscript (lines 213-217).

[Figure]

Figure R1: Proportions of the three qualities of fracture plane per rutschblock score
($N$ = 581).

- *I would suggest reversing the x-axis in the figure in the appendix (have very poor to the left and good to the right) to align with the other figures in the manuscript, and to insert this figure (as well as its results) in the results chapter. It could also be beneficial to add a paragraph discussing, comparing or summarising the results of the three different scales*

*used for stability (categorising stability into 3, 4 and 5 classes)*

We changed the x-axis as suggested. As we provided the figure in the Appendix primarily for illustration and to facilitate the comparison to the previous study by Techel et al. (2020), we prefer to keep it in the Appendix.

*Technical corrections*
*Now follow specific comments, with reference to line numbers in the manuscript:*

- *#30-37 This is a description of a regional forecast, I suggest you add the word "regional" somewhere, and briefly describes how regional warnings are different from other forecasts (object-based / slope-scale) and, if possible, if this study may be valuable also to slope-scale assessments/forecasts.*

We now explicitly refer to the regional scale in the revised manuscript (line 35) and endangered objects (line 36). For assessing the risk while skiing a single slope or the risk to infrastructure, the local conditions (release probability and consequences (including avalanche size) need to be considered rather than the regional forecast (lines 36-37).

- *#32 I could not easily find the EAWS 2019 in the reference list (it is on #570)*

Thanks for pointing this out. We removed this inconsistency in the revised manuscript.

- *#39 Could you improve the English of the sentence starting with "Even"?*

We revised the sentence.

- *#59-60 Consider rewrite and simplify the first sentence, e.g. "but it was not possible to distinguish 2–Moderate from 1–Low"*

We revised the sentence as suggested.

- *#84 Explain why you selected dry snow conditions only (dry slab selection could also be explained in #113)*

Stability evaluation for dry-snow conditions is of particular relevance to mitigate the persistent weak layer avalanche problem type. Profile data are primarily indicative of dry-snow conditions. Also, signs of instability such as whumpfs and shooting cracks relate to dry-snow slab avalanches. Hence the data we analyse are simply not suited for other than dry-snow conditions. We now provide a reason why we focus on dry-snow conditions (lines 88-89).

- *#104 Could you improve the definition of the "adjacent layer" and explain when it is*

*above and below the failure interface?*

We followed the approach that was introduced by Schweizer and Jamieson (2003). In all profiles, the failure interface was reported. While in most cases the failure layer was obvious, we considered the softer of the two layers as the failure layer (FL) and the layer across the failure interface as the adjacent layer (AL). If there was no difference in hardness, we selected the lower layer as the failure layer, and the layer above the failure interface as the adjacent layer. We have now clarified this procedure in the revised manuscript (lines 110-114).

*- #108 Explain why you derived three stability classes her, while using five classes above (#95) and even 4 classes in the appendix*

Traditionally, five classes were used as described by Schweizer and Wiesinger (2001). In various subsequent studies simplifications were sought, often to ease interpretation or to make it amenable for automatization. Along these lines, Schweizer et al. (2008) proposed to consider RB score, RB release type and the threshold sum to assess stability. This can be seen as a simplification of the 5-class system; moreover, it is amenable to automatization. The 4-class system is based on RB score and RB release type only, again to further automated classification. There is no single best classification scheme as long stability cannot be reliably measured.

*- #171-174 Please simplify this description*

We revised the description and deleted some parts.

*- #242 Please explain the acronym "RF" (presumably regional forecast)*

We now introduce RF and LN at the very beginning in the Data and Methods section of the revised manuscript (lines 84-86).

*- #245 Please clarify: Did the observer estimate the LN for the same region / area as the RF or is LN valid for a smaller area than RF and thus probably biased towards lower danger levels (presuming that RF is the highest danger level in the region)?*

Many thanks for pointing out this scale issue, which we now discuss it in the revised manuscript (lines 462-463). We also provided a figure in the Reply to Reviewer #2 that illustrates this issue (Figure R2). Moreover, we now clarify that the assessments refer to two different scales (lines 84-86).

*- #277 I presume "even" could be deleted (or is the sentence incomplete?)*

We revised this sentence.

*-	#367 Could you improve the sentence "the split variables and values were plausible"?*

We improved this sentence.

*-	#436 Could you improve the sentence "only in 1 out 5 cases differences in snow depth were indicative of snow instability"?*

We revised this sentence.

*-	#523 Please explain how/why this poses a challenge to forecasting?*

Persistent weak layers can linger for a long time in the snowpack and represent the failure layer in skier-triggered avalanches. Hence, risk assessment and forecasting is challenged by the presence of persistent weak layers. Moreover, it is challenging to communicate a danger (or the avalanche problem type: persistent weak layer) that is "invisible" and often only sporadically present (line 555).

**References**

Schweizer, J., and Wiesinger, T.: Snow profile interpretation for stability evaluation, Cold Reg. Sci. Technol., 33, 179-188, https://doi.org/10.1016/S0165-232X(01)00036-2, 2001.

Schweizer, J., and Jamieson, J. B.: Snowpack properties for snow profile analysis, Cold Reg. Sci. Technol., 37, 233-241, https://doi.org/10.1016/S0165-232X(03)00067-3, 2003.

Schweizer, J., McCammon, I., and Jamieson, J. B.: Snowpack observations and fracture concepts for skier-triggering of dry-snow slab avalanches, Cold Reg. Sci. Technol., 51, 112-121, https://doi.org/10.1016/j.coldregions.2007.04.019, 2008.

Schweizer, J., Mitterer, C., Techel, F., Stoffel, A., and Reuter, B.: On the relation between avalanche occurrence and avalanche danger level, Cryosphere, 14, 737-750, https://doi.org/10.5194/tc-14-737-2020, 2020.

Techel, F., Müller, K., and Schweizer, J.: On the importance of snowpack stability, the frequency distribution of snowpack stability, and avalanche size in assessing the avalanche danger level, Cryosphere, 14, 3503-3521, https://doi.org/10.5194/tc-14-3503-2020, 2020.

**Reply to Reviewer #2**

We thank the reviewer, Dr. Birkeland, for his insightful comments. Below we provide answers to the major points and indicate how we will improve the manuscript.

*I selected "Major Revisions" for this paper simply because some of my comments suggest that the paper may benefit from re-analyzing some of the data. However, I do not think that such a re-analysis should be overly difficult, and so my review probably falls somewhere between "major revisions" and "minor revisions".*
*This research aims "to characterize the avalanche danger levels based on expert field observations of snow instability." This is an important goal and is something that would be beneficial for avalanche forecasters and for a better understanding avalanche danger. The authors utilize an interesting, unique, and high-quality data set. While I believe the authors have produced an interesting paper, I believe it would benefit from some changes and clarifications. I have four major comments:*

*1.        First, when I was initially and quickly reading the paper and looking at the title, I assumed that the "avalanche danger" referred to in the title was the regional avalanche danger rating. However, this is not the case. Rather the "avalanche danger rating" is really the local nowcast provided by the observer. As pointed out in the paper, there is a scale mismatch between a local rating for a particular area and a regional avalanche danger rating. In addition, the authors point out that they are utilizing a local nowcast in their abstract. Despite this, I believe the authors should more clearly define the differences between these two ratings and if they decide to utilize the local nowcast avalanche danger then that should be specified in the title.*

We regret if we did not clearly introduce the regional forecast vs. the local nowcast. We refer to regional forecast and local nowcast as introduced by Techel and Schweizer (2017). The regional forecast is the danger rating issued in the public bulletin. The local nowcast is the assessment by experienced forecasters, researcher or observers after a day of travelling in the backcountry. We relate local observations to the local nowcast, hence the avalanche danger as described for an area of about 100 km$^2$. By doing so, the scales match.
We now more clearly make the difference between local nowcast and regional forecast (e.g. lines 48-49, and lines 84-86). We also mention clearly in the Abstract that we refer to the local danger assessment described by the danger level.
We think the avalanche danger level summarizes the avalanche conditions regardless of the type of assessment (regional forecast vs. local nowcast). Therefore, we do not see the reason for changing the title and prefer to keep it as is.

*2.        Second, along the lines of my first comment, the authors acknowledge that there is circular reasoning in their data since the observers are making snowpack and avalanche observations and are also assigning the local avalanche danger. Undoubtedly the observers are taking their observations into account when they are assigning the local avalanche danger. I believe this is somewhat problematic. In this scenario the authors may actually be testing what snowpack observations the observers happen to associate with a particular local avalanche danger rating rather than the more general question of which snowpack observations are associated with a danger level. I am wondering why the authors didn't simply compare the local observations to the regional avalanche danger assigned for that region for*

*that day? That way there would be independence between the snowpack observer and the assessment of the avalanche danger, and the results would reflect the frequency of making these observations for a given regional danger level rating. I would suggest either utilizing the regional danger ratings or providing a solid rationale for not using them in the paper.*

We use the local nowcast since it is clearly a better descriptor of the avalanche conditions (Bakermans et al., 2010; Jamieson et al., 2009; Techel and Schweizer, 2017). The two main reasons are: (1) The local nowcast assessment is done after the regional forecast assessment and is no longer a forecast, hence it does not include forecast errors due to, for instance, errors in the weather forecast. (2) There is a scale mismatch between the local observations we relate the danger level to and the regional forecast. The regional forecast is by definition broader and cannot take into account peculiarities within the region. The regional forecast has to address the highest danger prevailing in the region. Hence, it is well possible that in some subregions the danger is actually lower (Figure R2). This danger assessment (local nowcast) should then be related to the local observations. That's the approach we follow, which is grounded in a tradition of forecasting inherently connected to verification.

[Figure]

**Figure R2:** Effect of the spatial resolution of a forecast (or the size of forecast region) on danger assessment and quality. In the upper square, showing a forecast domain, the expected avalanche conditions represented by a higher (dark red) and a lower (light blue) danger level (RF) are shown. This situation will be assessed and communicated differently, depending on the size of the warning regions used by the warning service (12 vs. 1 region). The circles represent local danger level estimates (LN) (adapted from Figure 6.2 in Techel, 2020).

Therefore, we are not convinced that it is advantageous to use the regional forecast. In our view, it is imperative to use the verification data, or at least the best possible assessment. This is obviously, by definition, not the forecast. While full independence is certainly desirable, we think it is almost impossible to achieve in the context of avalanche forecasting. Also, the observers are biased by the forecast. While the local nowcast has the disadvantage of introducing a potential bias, which we openly discuss, the regional forecast is obviously incorrect on 1-2 days per week and does not match the scale of observation. We think this obvious inaccuracy and the scale mismatch is more severe than the potential bias introduced by the local nowcast. Therefore, we prefer to relate the observations to the local nowcast.

We added the above reasoning why we use the local nowcast to the Discussion section (lines 470-476).

In addition, as a reaction on the reviewer's comment, we have explored an approach where we used the regional forecast (RF) as starting point for the local assessment. The corresponding classification trees are dominated by RF when predicting LN; the local observations rarely show up in the tree. The only exception is in the case when RF = 3 and there are no signs of instability, then the tree predicts LN = 2. In all other branches LN is predicted solely by RF. This is not surprising given the still relatively high agreement rate between RF and LN. However, this approach does not provide any guidance on how to assess the local danger level based on field observations, which is what we actually aim for.

*3.      Third, this paper utilizes a unique dataset of snow profiles and observations from Switzerland. Approximately 95% of the data are from the region of Davos. Later the authors explain that some of their results, such as the predominance of persistent weak layers, may be because so much of their data are from the Davos area. I believe the paper would benefit from using only those snow profiles and observations from around Davos, rather than a highly unbalanced dataset consisting of almost all profiles from Davos and then 5% from other areas. This would still retain 95% of the data but would remove some of the variability introduced by the other 5% of the data.*

We admit that at first glance the selection seems questionable with regard to geographical origin. However, we initially did not care about origin, but selected the profiles purely based on quality. We have now assessed whether those 5 % of profiles that were not recorded in the region of Davos, introduce any bias. As shown below (Figure R3, Table R1) this seems not to be the case. The profile characteristics are very similar. In particular snow depth, a good indicator of snow climate, is not significantly different (median snow depth 120 cm vs. 108 cm; $p$ = 0.13). Also, the snowpack structure index, which relates to the importance of faceting, is not different either (1.78 vs. 1.70; $p$ = 0.7).

[Figure]

**Figure R3:** Comparison of selected profile characteristics from the region of Davos ($N$ = 561) and from other regions (Rest, $N$ = 28): (a) Snow depth and (b) snowpack structure index. For median values and $p$-value see Table R1.

At first glance, these results may seem surprising. However, these are fully in line with the observation that in our study many snowpack characteristics were similar to those reported in previous studies that included profiles that originated from regions with clearly different snow

climates such as the Columbia mountains of western Canada. For instance, in the study by Schweizer and Jamieson (2003) profiles from the Swiss Alps and the Columbia Mountains were jointly analysed. While there were some differences between the profiles from the different snow climates (e.g. the frequency of occurrence of surface hoar), the characteristics of instability were largely the same in both sets of profiles. This suggests that characteristics of instability may well be similar in different snow climates as long as dry-snow problem types are relevant. Of course, the frequency of occurrence of these characteristics may still differ.

**Table R1**: Comparison of selected profile characteristics from the region of Davos ($N = 561$) and the other regions ($N = 28$). Median is shown (mode in the case of FL grain type) and level of significance based on non-parametric Mann-Whitney U-test.

| Parameter | Rest | Davos | *p* |
|---|---|---|---|
| Elevation (m a.s.l.) | 2488 | 2470 | 0.367 |
| Slope angle (°) | 34 | 33 | 0.301 |
| Snow depth (cm) | 120 | 108 | 0.131 |
| Slab thickness (cm) | 41 | 38 | 0.052 |
| FL grain size, avg (mm) | 1 | 1 | 0.643 |
| FL grain size, max (mm) | 1.5 | 1.5 | 0.846 |
| FL grain type | facets | facets | 0.872 |
| RB score | 4-5 | 4 | 0.737 |
| 5-class stability | fair | fair | 0.761 |
| $SNPK_{index}$ | 1.78 | 1.70 | 0.702 |

We now provide this information in Appendix C.

*4.        My fourth comment relates back to my third. The authors spend considerable effort (and content in their paper) characterizing the snow profiles. This provides some interesting results that I believe should be retained, but it is outside the primary stated goal of the paper. As stated above, I believe this analysis would have more meaning if the data were restricted to just the 95% of data from the region of Davos. Then the snowpack characterization part of the paper can provide a characterization of the Davos area snowpack rather than "mostly" the Davos area snowpack with 5% of the profiles and observations from other areas. In addition, this part of the paper should be better highlighted in the abstract and perhaps the title as well since nearly as much attention is paid to this snowpack characterization as is paid to the relationship of the snow profiles and observations to the danger levels. I think that this "characterization of a snowpack" in a region is quite valuable and will set a baseline for future work which could compare this characterization against the characterization of the snowpack in other regions or other countries.*

We thank the reviewer for his favourable assessment of the part on snowpack characterization. As shown above, the variability introduced by those 5 % of profiles is not really significant. Hence our conclusion that our results reflect the snow climate of Davos and might be of limited value for other regions is actually not fully supported and seems too restrictive. Given the insignificant bias introduced by those 5 % of profiles, we see little advantage in re-doing the analyses. We now provide the details on the differences between the two groups in Appendix C.

*While I do have some substantial comments that I believe the authors should address, I do think that this is important work and that it should be published following revisions.*
*In addition to the above comments, I have some more minor comments and suggested typographical corrections:*

*Line 29: Delete "at times"*
*Line 40: Delete "were" and replace with "have been"*
*Line 68: Replace "inexistent" with "nonexistent"*

Many thanks; we made these changes as suggested.

*Line 84: Can you provide more specifics about how you defined "an experienced observer"?*

Experienced observers are those who have done dozens of high-quality profiles. They are almost exclusively (96 %) professional forecasters or researchers with extensive experience in field work. The authors recorded 63 % of all profiles.

*Line 97: After reading this paragraph, I am still not certain how the failure layer and adjacent layers are defined. I understand how the authors come up with the failure interface, but how do they necessarily define the failure layer? Was this done manually by the authors? And, with the adjacent layer, was this typically just the layer adjacent to the failure interface that was not the failure layer? I think this might be less confusing if the authors talked about the failure interface, and then the layer above that interface and the layer below that interface. They could then also quantify how often the "failure layer" is below or above the layer interface.*

We regret that the description was not clear enough. We follow the approach that was introduced by Schweizer and Jamieson (2003). In all profiles, the failure interface was reported. While in most cases the failure or weak layer is obvious, we considered the softer of the two layers as the failure layer (FL) and the layer across the failure interface as the adjacent layer. If there was no difference in hardness, we selected the lower layer as the failure layer, and the layer above the failure interface as the adjacent layer.
We now provide more details on the procedure of failure layer selection (lines 110-114).
We already provided the failure location with regard to the failure interface (above vs. below) for failure layers with persistent and non-persistent grain types individually. We now added that, overall, the failure layer was in 53 % below the failure interface, and in the remaining 47 % above (line 181).

*Line 115 – It seems like assigning 3+ to Considerable is arbitrary. Why not apply High here since 2+ is also assigned to Considerable?*

We regret that the description was not clear enough. We assumed that + and - refer to a somewhat higher and lower danger level, respectively. Hence, we assigned 2-, 2, and 2+ to *2–Moderate*, and 3-, 3, and 3+ to *3–Considerable*. Occasionally, intermediate values were given, indicated with "2 to 3", those we assigned the next higher danger level, therefore, *3–*

*Considerable* in this example. Overall, intermediate values were provided in only 14 % of the cases. We now provide this information in the revised manuscript (line 124).

*Line 199-200 and line 525 – I don't think that a decrease in the whole block release of RBs with increasing RB number is necessarily related to a decrease in crack propagation propensity. This may have more to do with the increased damage to the slab caused by harder jumping on the RB that causes partial block releases. Given the complexities involved, I don't think the authors can draw such a definitive conclusion from their data. While I don't have a good dataset to either confirm or refute this conclusion, I have anecdotally seen seasons where PSTs consistently propagate to end – indicating the propensity for crack propagation – for a long time after other tests have indicated that failure initiation is far less likely.*

We agree with the reviewer's experience of full propagation in PST tests; we certainly made similar observations. However, we also have datasets showing that shear strength as well as specific fracture energy of the weak layer increase with time. This suggests that initiation as well as propagation become less likely with time – in line with our suggestion. Our interpretation is simply a suggestion, clearly not a firm conclusion, and may hopefully trigger some further research.
We re-worded the statements to better express the fact that it is a possible interpretation (lines 210-211 and line 557).

*Line 454: Delete "and" and replace with "or"*

Thanks, changed as suggested.

*Line 528: I assume the "avalanche danger level" in the sentence on this line is the local avalanche danger? As stated in one of my primary comments, it would be helpful to make sure to careful differentiate between the local avalanche danger rating and the "avalanche danger rating". I always assume this latter term is associated with the regional avalanche danger rating.*

We now refer to the nowcast in the revised manuscript as suggested (lines 560-561). As mentioned above, we assume that the danger rating describes an avalanche situation regardless of whether this assessment is made in the office, in the field, beforehand or in hindsight.

*Line 529: Along the lines of my comment above, the authors state that the local avalanche danger rating agrees with 70% of the regional danger ratings. It would be interesting to more thoroughly compare how those two ratings at the two different scales differ.*

We agree that this would be interesting. In fact, Techel and Schweizer (2017) provided a detailed comparison between regional forecast and local nowcast. This topic, essentially verification of forecast, is beyond the scope of the present study. Nevertheless, Figure R4 provides the agreement by danger level. The agreement was 95 %, 66 %, 71 % and 33 % for

the danger levels *1–Low*, *2–Moderate*, *3–Considerable* and *4–High*, respectively. We now provide Figure R4 in Appendix B.

[Figure]

**Figure R4:** Deviation between regional forecast and local nowcast for each of the four danger levels (a-d) from *1–Low to 4–High* (*N* = 583).

**References**

Bakermans, L., Jamieson, B., Schweizer, J., and Haegeli, P.: Using stability tests and regional avalanche danger to estimate the local avalanche danger, Ann. Glaciol., 51, 176-186, https://doi.org/10.3189/172756410791386616, 2010.

Jamieson, B., Haegeli, P., and Schweizer, J.: Field observations for estimating the local avalanche danger in the Columbia Mountains of Canada, Cold Reg. Sci. Technol., 58, 84-91, https://doi.org/10.1016/j.coldregions.2009.03.005, 2009.

Schweizer, J., and Jamieson, J. B.: Snowpack properties for snow profile analysis, Cold Reg. Sci. Technol., 37, 233-241, https://doi.org/10.1016/S0165-232X(03)00067-3, 2003.

Techel, F., and Schweizer, J.: On using local avalanche danger level estimates for regional forecast verification, Cold Reg. Sci. Technol., 144, 52-62, https://doi.org/10.1016/j.coldregions.2017.07.012, 2017.

Techel, F.: On consistency and quality in public avalanche forecasting - a data-driven approach to forecast verification and to refining definitions of avalanche danger, Ph.D., Faculty of Science, University of Zurich, Zurich, Switzerland, 236 pp., 2020.

---

## Author Response (AR2)

**Reply to Editor's comments**

Dear Editor

Many thanks for your assessment of our manuscript.

We appreciate the language edits and have accepted them all as suggested.

Moreover, we now provide the details of what we consider an "experienced" observers in the manuscript (lines 91-93).

Best regards,

Jürg Schweizer.

---

## Author Response (AR3)

**Reply**

As far as I understand the paper is accepted and no reply is needed.

Best regards,

Jürg Schweizer.